# Machine learning and biological validation identify sphingolipids as potential mediators of paclitaxel-induced neuropathy in cancer patients

Jörn Lötsch[1,2†], Khayal Gasimli[3†], Sebastian Malkusch[1,2], Lisa Hahnefeld[1,2], Carlo Angioni[1], Yannick Schreiber[2], Sandra Trautmann[1,2], Saskia Wedel[1], Dominique Thomas[1,2], Nerea Ferreiros Bouzas[1,2], Christian H Brandts[4,5], Benjamin Schnappauf[6], Christine Solbach[3], Gerd Geisslinger[1,2], Marco Sisignano[1,2]*

[1]Institute of Clinical Pharmacology, Goethe - University, Frankfurt, Germany; [2]Fraunhofer Institute for Translational Medicine and Pharmacology ITMP, and Fraunhofer Cluster of Excellence for Immune Mediated Diseases CIMD, Frankfurt, Germany; [3]Goethe University, Department of Gynecology and Obstetrics, Frankfurt, Germany; [4]German Cancer Consortium (DKTK) and German Cancer Research Center (DKFZ), Heidelberg, Germany; [5]Goethe University, University Cancer Center Frankfurt (UCT), Goethe University Hospital, Frankfurt, Germany; [6]Oncology Center, Sana-Klinikum Offenbach, Starkenburgring, Germany

*For correspondence:
Marco.Sisignano@med.uni-frankfurt.de

†These authors contributed equally to this work

Competing interest: The authors declare that no competing interests exist.

## Abstract

**Background:** Chemotherapy-induced peripheral neuropathy (CIPN) is a serious therapy-limiting side effect of commonly used anticancer drugs. Previous studies suggest that lipids may play a role in CIPN. Therefore, the present study aimed to identify the particular types of lipids that are regulated as a consequence of paclitaxel administration and may be associated with the occurrence of post-therapeutic neuropathy.

**Methods:** High-resolution mass spectrometry lipidomics was applied to quantify d=255 different lipid mediators in the blood of n=31 patients drawn before and after paclitaxel therapy for breast cancer treatment. A variety of supervised statistical and machine-learning methods was applied to identify lipids that were regulated during paclitaxel therapy or differed among patients with and without post-therapeutic neuropathy.

**Results:** Twenty-seven lipids were identified that carried relevant information to train machine learning algorithms to identify, in new cases, whether a blood sample was drawn before or after paclitaxel therapy with a median balanced accuracy of up to 90%. One of the top hits, sphinganine-1-phosphate (SA1P), was found to induce calcium transients in sensory neurons via the transient receptor potential vanilloid 1 (TRPV1) channel and sphingosine-1-phosphate receptors.SA1P also showed different blood concentrations between patients with and without neuropathy.

**Conclusions:** Present findings suggest a role for sphinganine-1-phosphate in paclitaxel-induced biological changes associated with neuropathic side effects. The identified SA1P, through its receptors, may provide a potential drug target for co-therapy with paclitaxel to reduce one of its major and therapy-limiting side effects.

**Funding:** This work was supported by the Deutsche Forschungsgemeinschaft (German Research Foundation, DFG, Grants SFB1039 A09 and Z01) and by the Fraunhofer Foundation Project: Neuropathic Pain as well as the Fraunhofer Cluster of Excellence for Immune-Mediated Diseases (CIMD). This work was also supported by the Leistungszentrum Innovative Therapeutics (TheraNova) funded

by the Fraunhofer Society and the Hessian Ministry of Science and Arts. Jörn Lötsch was supported by the Deutsche Forschungsgemeinschaft (DFG LO 612/16-1).

## eLife assessment

Sisigano et al. report findings about the role of sphingolipids using lipidomics with machine learning in paclitaxel-induced peripheral neuropathy and preliminary translation of the impact of SA1P in cultured neuronal cells. This study presents a **valuable** finding on the increased activity of two well-studied signal transduction pathways in a subtype of breast cancer. The significance is limited by **incomplete** evidence which can be addressed in larger clinical cohorts in the future and with more robust biological validation approaches.

## Introduction

Paclitaxel is a standard adjuvant treatment for breast cancer and several other cancers. Originally isolated from the yew tree *Taxus brevifolia*, it inhibits mitosis by stabilizing microtubules and preventing tubulin depolymerization (*Gornstein and Schwarz, 2014*; *Hershman et al., 2014*; *Yang and Horwitz, 2017*). A serious dose- and therapy-limiting side effect, which it shares with other commonly used cytostatic drugs, is the chemotherapy-induced peripheral neuropathy and neuropathic pain (CIPN), which affects up to 80% of treated patients (*Pachman et al., 2011*; *Park et al., 2013*). Patients report a variety of primarily sensory symptoms, encompassing sensations like numbness, paresthesia, spontaneous pain, and heightened sensitivity to mechanical and/or cold stimuli in their hands and feet. In more severe instances, the loss of vibration sense and joint position sense can further impact their functionality (*Cavaletti and Marmiroli, 2010*). Engaging in essential daily activities becomes challenging for patients, leading to difficulties in tasks like fine finger movements (such as buttoning clothing). Walking can induce pain due to mechanical hypersensitivity, while handling tasks like retrieving items from a fridge or exposure to cold weather may exacerbate symptoms (cold hypersensitivity). Chemotherapy can also emerge post-treatment in a state termed 'coasting' where mild neuropathy can worsen, or new instances of CIPN may develop (*Flatters et al., 2017*). Currently, there are no pharmacologic treatments for CIPN expert for duloxetine (*Pachman et al., 2011*; *Cavaletti and Marmiroli, 2020*; *Smith et al., 2013*). Therefore, research on the mechanism of paclitaxel induced CIPN with possible identification of novel treatments is an active research topic.

Several genes and neurofilament proteins have been implicated in paclitaxel-induced neuropathy (*Diaz et al., 2018*; *Sisignano et al., 2019*; *Huehnchen et al., 2022*). More recently, lipid mediators have been shown to be produced at high levels in sensory neurons, neuronal tissue and immune cells after chemotherapy due to oxidative stress and have been shown to contribute to chemotherapy-induced neuropathy and neuropathic pain by modulating neuronal ion channels (*Hohmann et al., 2017*; *Piomelli and Sasso, 2014*; *Sisignano et al., 2016*; *Stockstill et al., 2018*). Therefore, they are of particular interest as signaling molecules and potential markers for chemotherapy-induced neuropathy in patients. In fact, lipids are already considered markers for other neurological diseases such as Alzheimer's disease and amyotrophic lateral sclerosis (ALS; *Jensen et al., 2020*; *Chen et al., 2018*; *Igarashi et al., 2011*; *Kim et al., 2017*). However, a systematic approach to investigate lipids in patients with paclitaxel-induced neuropathy has not been performed.

In this prospective clinical cohort study, plasma concentrations of 255 different lipid mediators were evaluated for changes in their concentrations associated with paclitaxel treatment. A comprehensive LC-MS/MS-based targeted and LC-QTOFMS-based untargeted lipidomics screening was performed on plasma samples from paclitaxel-treated patients. Lipid groups measured included eicosanoids, endocannabinoids, oxidized linoleic acid metabolites, sphingolipids, lysophospholipids and free fatty acids, many of which have been previously associated with persistent pain states (*Osthues and Sisignano, 2019*; *Shapiro et al., 2016*; *Sisignano et al., 2014*). A data-driven approach was used to identify lipid mediators whose concentrations could be used to train machine learning algorithms to identify, in new cases, whether a plasma sample was collected before or after therapy, or from a patient with or without post-therapy neuropathy. The biological relevance of the findings was then validated in vitro by applying SA1P to primary sensory neurons using calcium imaging.

# Materials and methods

## Patients and study design

This was a prospective single-arm study enrolling patients with breast cancer. The study was conducted in accordance with the Declaration of Helsinki on Biomedical Research Involving Human Subjects and was approved by the Ethics Committee of the Medical Faculty of the Goethe-University, Frankfurt am Main, Germany (reference number 4/09). Informed written consent was obtained from each of the participants.

Sixty patients (one male, 59 females) with breast cancer and undergoing paclitaxel treatment were recruited from the Tumor Center of the University Hospital Frankfurt, Germany (UCT). Most patients (n=47) received the 'paclitaxel-weekly' schedule, consisting of 12 cycles of paclitaxel treatment (80 mg/m², each week); a few patients (n=13) received mixed carboplatin/paclitaxel treatment (**Supplementary file 1**). A blood sample was collected from each patient before and after chemotherapy, and the degree of neuropathy after chemotherapy was assessed as described below. Plasma was isolated from the blood samples immediately after blood collection to ensure lipid stability. Plasma was stored at –80 °C until analysis.

All patients provided a blood sample before chemotherapy; however, only 36 patients of the patients provided a second blood sample after chemotherapy. For our analysis, we focused on patients that had two blood samples. Therefore, 72 samples from 36 patients (before and after chemotherapy) were analyzed using both LC-MS/MS-based targeted and LC-QTOFMS-based untargeted lipidomics. A total of 255 individual lipids were detected in each sample. From the resulting data, five patients had to be excluded due to incomplete lipidomics data (more than 20% of the analytes could not be detected). The remaining 62 samples from 31 patients were used for the machine learning analysis (**Figure 1**).

## Assessment of neuropathy

The occurrence and severity of peripheral neuropathy was assessed according to the guidelines of the NCI Common Terminology Criteria for Adverse Events (CTCAE) v5.0, Published: November 27, 2017, by the U.S. Department of Health and Human Services. Neuropathy assessment was performed prospectively, that is before the first paclitaxel treatment and again after the 12th treatment cycle. Neuropathy was assessed regularly upon visit of the patient. The last assessment was performed 4.5 years after initial chemotherapy (**Supplementary file 1**).

The severity of neuropathy was graded into five grades: Grade 1 Mild; asymptomatic or mild symptoms; clinical or diagnostic observations only; intervention not indicated. Grade 2 Moderate; minimal, local or noninvasive intervention indicated; limiting age-appropriate instrumental ADL (activities of daily living). Grade 3 Severe or medically significant but not immediately life-threatening hospitalization or prolongation of hospitalization indicated; disabling; limiting self-care ADL. Grade 4 Life-threatening consequences; urgent intervention indicated. Grade 5 Death related to AE. In the present cohort, grades 1–3 were detected following paclitaxel chemotherapy (**Supplementary file 1**). Of the 31 patients with a full set of samples, 17 had neuropathy after chemotherapy (54.9%), 12 had grade 1, 3 had grade 2 and two patients with grade 3 (**Supplementary file 1**).

## Lipidomics analysis using LC-MS/MS and LC-QTOFMS

Blood was collected from patients in EDTA tubes and immediately centrifuged at 2000 x *g* for 10 min at 4 °C.The supernatant was immediately frozen at –80 °C until further processing. Approximately 2–3 ml of plasma was considered sufficient for each patient. Liquid chromatography-tandem mass spectrometry (LC-MS/MS) analysis of eicosanoids oxidized linoleic acid metabolites (O(x)LAMs), prostanoids, endocannabinoids, LPAs, pterins, sphingolipids and ceramides, and lipidomics screening were performed as described previously (**Sisignano et al., 2016**; **Shapiro et al., 2016**; **Brunkhorst-Kanaan et al., 2019**). A total of 255 lipids were quantified in each plasma sample. These lipids belong to the groups of eicosanoids, oxidized linoleic acid metabolites, endocannabinoids, lysophosphatidic acids, pterins, sphingolipids, ceramides, cholesterols, cholesterol esters, diacylglycerols, triacylglycerols, phospholipids, lysophospholipids, and free fatty acids. Full details of the lipids detected are provided in **Supplementary file 2**.

Lipid extraction was performed using different solvents and methods based on the analyte group. Arachidonic acid, linoleic acid, eicosanoids, O(x)LAMs, and prostanoids were extracted using

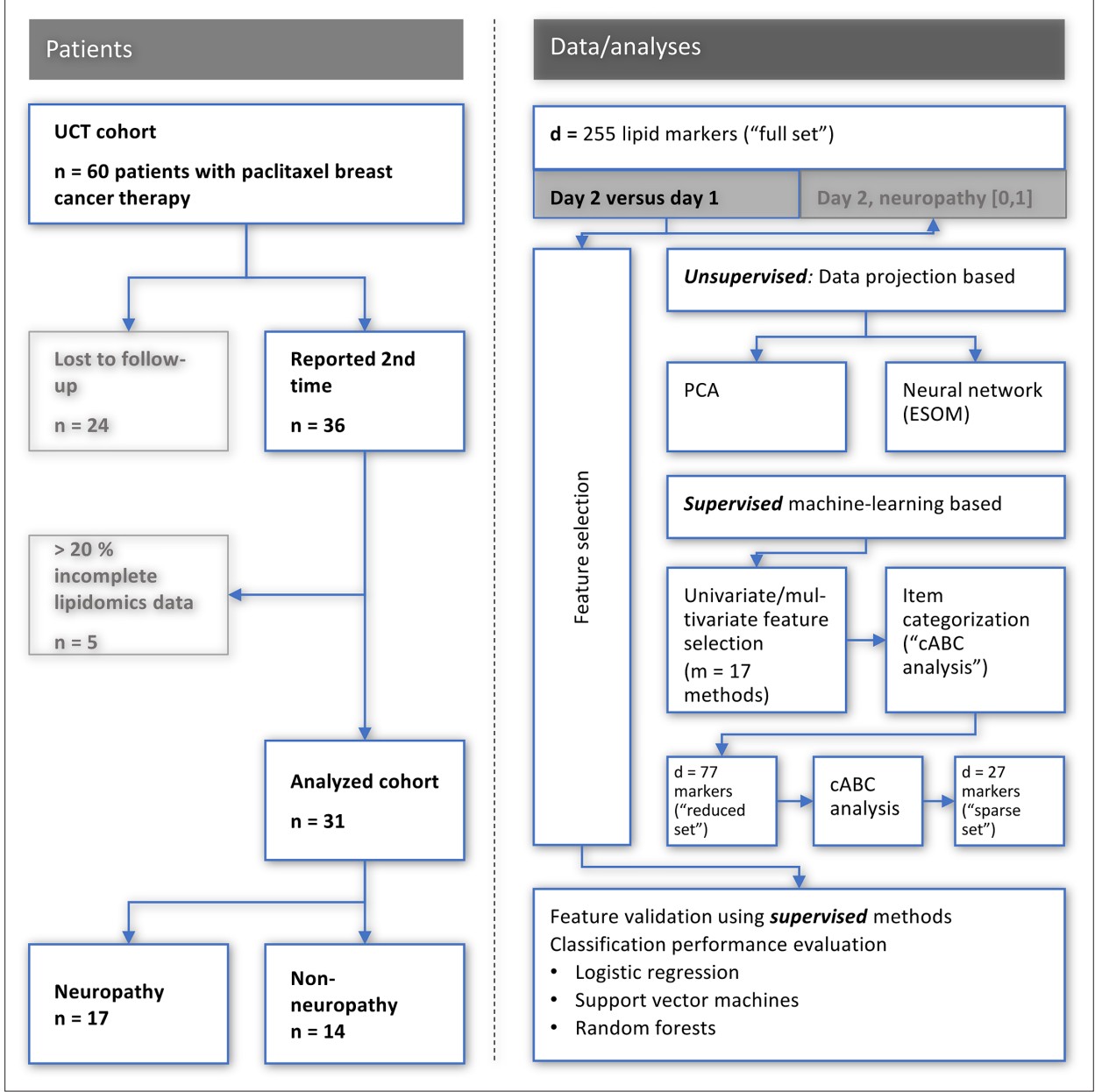

**Figure 1.** Flowchart showing the number of patients included and the workflow of the data analysis. UCT: University Cancer Center Frankfurt, PCA: principal component analysis, ESO: emergent self-organizing maps, cABC analysis: computed ABC analysis. The figure was created using Microsoft PowerPoint (Redmond, WA, USA) on Microsoft Windows 11 running in a virtual machine powered by VirtualBox 6.1.36 (Oracle Corporation, Austin, TX, USA) as guest on Linux, and then further modified with the free vector graphics editor "Inkscape version 1.2 for Linux, https://inkscape.org/.

a liquid-liquid extraction method with ethyl acetate as the solvent. Endocannabinoids were also extracted using liquid-liquid extraction but with a mixture of ethyl acetate and hexane. Lysophosphatidic acids (LPA) were extracted using disodium hydrogen phosphate/citrate buffer and butanol. Pterins were extracted using solid-phase extraction (SPE) with Oasis MCX 96-Well plates after preincubation with a 1 M HCl-iodine-potassium iodide solution. Sphingolipids and untargeted lipidomics were extracted using liquid-liquid extraction with methanol/chloroform and methanol/methyl-tert-butyl-ether, respectively.

Arachidonic acid and linoleic acid were measured using 10 µl sample volume on an Agilent 1200 LC system with a Gemini NX C18 column and precolumn, coupled to a QTrap 5500 MS system in negative ionization mode. Eicosanoids and O(x)LAMs were measured using 200 µl sample volume on the same LC and MS systems. Prostanoids and endocannabinoids were measured using 200 µl and

100 µl sample volumes, respectively, on an Agilent 1290 Infinity LC system with an Acquity UPLC BEH C18 column and precolumn, coupled to a QTrap 6500+MS system in negative and positive ionization modes, respectively. Lysophosphatidic acids (LPA) were measured using 100 µl sample volume on an Agilent 1200 LC system with a Luna C18 column and precolumn, coupled to a QTrap 5500 MS system in negative ionization mode. Pterins were measured using 50 µl sample volume on an Agilent 1200 LC system with a Synergi Hydro RP 2.0x250 mm column, coupled to a QTrap 5500 MS system in negative ionization mode. Sphingolipids were measured using 10 µl sample volume on an Agilent 1200 LC system with an additional 1100 isocratic pump and a Zorbax Eclipse Plus C18 column with a Luna C18 precolumn, coupled to a QTrap 5500 MS system in positive ionization mode. Untargeted lipidomics was measured using 20 µl sample volume on a Nexera X2 UHPLC system with a Zorbax Eclipse Plus C8 column and precolumn, coupled to a TripleTOF 6600 MS system in both negative and positive ionization modes.Data Acquisition of the QTrap systems was done using Analyst Software 1.6.3 and quantification was performed with MultiQuant Software 3.0.2 (both Sciex, Darmstadt, Germany), employing the internal standard method (isotope dilution mass spectrometry). Calibration curves were calculated by linear regression with $1/x$ weighting. Data acquisition of the TripleTOF was controlled by Analyst TF 1.7 software. Lipids were identified using exact mass (±5 ppm), isotope distribution and by MS/MS fragmentation pattern using MasterView 1.1 software and finally relatively quantified using MultiQuant 3.0.2 software (all Sciex, Darmstadt, Germany). Results were normalized to the first QC sample using the median peak ratio from MarkerView Software V1.21 (Sciex, Darmstadt, Germany) for positive and negative ionization mode, respectively.

## Data analysis

Programming was performed in the R language (*Ihaka and Gentleman, 1996*; version 4.1.2, for Linux), available free of charge from the Comprehensive R Archive Network (CRAN) at https://CRAN.R-project.org/ (*R Development Core Team, 2008*), and in the Python language (*Van Rossum and Drake, 1995*) (version 3.8.12), available at https://www.python.org (accessed March 1, 2022). The main packages used for R-based data analysis are given along with the description of the methods. The main packages used for the Python-based data analysis were the numerical Python package 'numpy' (https://numpy.org *Harris et al., 2020*), 'pandas' (https://pandas.pydata.org *McKinney, 2010*; *pandas-dev/pandas, 2020*), fundamental algorithms for scientific computing in Python 'SciPy' (https://scipy.org *Virtanen et al., 2020*) and 'scikit-learn' (https://scikit-learn.org/stable/ *Pedregosa et al., 2011*). Analyses were performed on an AMD Ryzen Threadripper 3970 X (Advanced Micro Devices, Inc, Santa Clara, CA, USA) computer running Ubuntu Linux 20.04.4 LTS (Canonical, London, UK).

The data analysis (*Figure 1*) combined statistical and machine learning methods in a 'mixture of experts' approach shown to be superior to single method designs (*Khadirnaikar et al., 2023*; *Hu et al., 1997*) such as regression analysis alone (*Leclercq et al., 2019*). The reason for the preference for using several but one method is that all statistical models inherently rely on underlying assumptions about a data set, some of which can be tested, but in practical scenarios it is often difficult to determine the best model to describe a real data set. All tests were two-sided, that is without a directed hypothesis, without specific expectations as to which of the d=255 lipid markers would be regulated as an effect of paclitaxel and/or as a sign of neuropathy, and in which direction this regulation might occur. Data visualization followed advisees of *Lötsch and Ultsch, 2023c*. The data analysis included unsupervised and supervised methods (*Figure 1*). The former were used to determine whether the lipidomics data contained structures that supported prior classifications into baseline versus post-treatment samples, or into subjects with or without post-treatment neuropathy. Supervised methods were then used to identify lipid mediators that carried information relevant to this class structure of the data set. The detailed analyses are described below.

## Data preprocessing

Data preprocessing included examination of the distribution of the variables including evaluation of possible transformations along Tukey's ladder of powers (*Tukey, 1977*; *Box and Cox, 1964*), supported by visualizing the data using quantile-quantile plots and assessing the normal distribution using D'Agostino and Pearson tests (*D'Agostino and Pearson, 1973*; *D'agostino, 1971*) implemented in the 'SciPy' Python package. This suggested logarithmic transformation. Imputation of

missing values was performed by means of random forests (*Ho, 1995*; *Breiman, 2001*) implemented in our R library 'pguIMP' (https://cran.r-project.org/package=pguIMP *Malkusch et al., 2021*). Only variables or cases with less than 20% missing values were retained.

## Unsupervised analysis of lipid markers to identify class structures

After log transformation and missing value imputation, lipid marker concentrations were analyzed using unsupervised methods to assess whether they contained structures consistent with a prior class structure, such as pre-/post-treatment samples or samples from patients with or without neuropathy. Z-standardized data were projected from the high-dimensional space onto lower dimensional planes by means of principal component analysis (PCA) (*Hotelling, 1933*; *Pearson, 1901*). However, among limitations of PCA is that it focuses on the dispersion (variance) of the data, while clustering/subgrouping attempts to identify concentrations (neighborhoods) within the data, making PCA and clustering opposing methods in this sense. Furthermore, PCA fails to separate data sets with non-linear relationships, for which other projection methods have been developed, such as the currently popular t-distributed stochastic neighborhood embedding (t-SNE) (*Van der Maaten and Hinton, 2008*), which also has limitations and occasionally provides class structure to data sets that do not have class structure (*Lötsch and Ultsch, 2019*). Considering these limitations, a second unsupervised approach was used to verify the agreement between the lipidomics data structure and the prior classification, using unsupervised machine-learning applied as self-organizing maps (SOM) of artificial neurons (*Kohonen, 1982*). Of note, as mentioned previously (*Lötsch and Ultsch, 2019*), there are two types of SOMs, with one using a small number of neurons that are identified with clusters and a second type of which one feature is the usage of a large number of neurons up to thousands (for details about the number of neurons, see *Lötsch et al., 2018a*), termed emergent SOM (ESOM), where emergence, that is the appearance of higher-level structures due to micro-scale interactions, can be observed by looking at structures like ridges or valleys consisting of groups of neurons (*Ultsch et al., 2007*). The latter was used in the present analysis, that is an emergent self-organizing map (ESOM) of artificial neurons, followed by visualization of class structure using a U-matrix (*Ultsch and Lötsch, 2017*; *Ultsch and Locarek-Junge, 2003*; *Ultsch, 2003*). In the present implementation, the network consisted of 4000 neurons arranged on a two-dimensional toroidal grid with 50 rows and 80 columns (*Ultsch and Lötsch, 2017*; *Ultsch, 2003*). Each neuron has a position vector on the grid and a vector containing 'weights' with the same dimensions as the input dimensions, which were the z-standardized lipid marker concentrations. The weights were first randomly drawn from the datasets and then fitted to the data during the 20-epoch learning phase. This resulted in a representation of the dataset instances on a two-dimensional toroidal map as localizations of the respective 'best matching units'. Furthermore, the distances between the data points were calculated using the so-called U-matrix (*Sieman HP, 1990*; *Lötsch and Ultsch, 2014*), where the 'height" represents the average high-dimensional distance of a prototype with respect to all immediately neighboring prototypes. The corresponding visualization technique uses a topographic map including coloring. These calculations were performed using our R package 'Umatrix' (https://cran.r-project.org/package=Umatrix *Lötsch et al., 2018a*).

## Supervised analysis to identify lipid mediators relevant to the class structure of the dataset

### Segregation of validation data

For the supervised machine learning approach, a validation subsample of 20% of cases from each subgroup was reserved for final model validation. This ensured a representative holdout set across categories, with all data points from selected participants completely separated from the main dataset. The validation subsample remained untouched during feature selection and classifier training, allowing for robust evaluation of machine learning model performance on unseen data and helping to avoid overfitting. The remaining 80% of the dataset was used for training and testing the models. Thus, the following analyses were performed on the 80% training/test subsample.

## Feature selection

To the relevant lipid mediators for the class structure, that is into pre- and posttreatment or neuropathy positive or negative after therapy, several feature selection methods (*Guyon, 2003*) were used. This used machine learning for knowledge discovery. The approach assumes that if a classifier can be trained to assign a patient to the correct class better than by guessing, then the features, that is the lipid mediators in the dataset needed by the classifier to accomplish this task, contain relevant information about the addressed class structure. In this way, the most informative lipids can be identified. In this use of feature selection, creating a powerful classifier is not the final goal, but feature selection takes precedence over classifier performance. This means that the analysis is considered as successful when the class assignment is just better than guessing and the variables needed for this assignment have been identified.

Specifically, applied feature selection methods included (i) PCA-based variable importance described above. Second, univariate feature selection methods were implemented as (ii) calculation of effect sizes expressed as (*Cohen, 1960*), (iii) based on the false-positive rate, (iv) based on the family-wise error rate, and as (v) selection of the *k* best variables based on F-statistics as implemented in the 'SelectKBest' method available in the 'sklearn.feature_selection' module of the scikit-learn Python package. There, the number *k* of features to be selected was determined by a grid search of [1,...,18] variables in analyses based on the models specified below. Further feature selection methods were based on assignment of an importance measure to each variable following training of classification algorithms. Specifically, support vector machines (SVM *Cortes and Vapnik, 1995*) and random forests (*Ho, 1995*; *Breiman, 2001*) were selected as two commonly used classification algorithms of different types, that is class separation using hyperplanes in data projected to higher dimensions, or class separation using an ensemble of simple decision trees (for an overview of machine algorithms suitable for pain-related data, see *Lötsch and Ultsch, 2018b*). In addition, logistic regression was included as a classical method for class assignment (*Cramer, 2002*). Following training of the algorithms, the most relevant variables were selected using methods available in the 'sklearn.feature_selection' module of scikit-learn, including (vi) 'SelectFromModel' (SFM), which selects features based on importance weights in the trained algorithm, and (vii) recursive feature elimination (RFE), which selects features, by recursively considering smaller and smaller feature sets and generating a feature ranking.

Algorithm-based feature selection was performed after 20% of the members of each class had been put aside as a validation sample, which was not further touched during algorithm training and feature selection. Subsequently, hyperparameter tuning was performed for the selected algorithms including using a fivefold cross validated grid search scenario as default in the 'GridSearchCV' method of the 'sklearn.model_selection' module of scikit-learn, during which also the penalty measure was chosen from the regularization methods (i) least absolute shrinkage and selection operator (LASSO) (*Santosa and Symes, 1986*; *Tibshirani, 1996*) or (ii) Ridge regression (*Hilt and Seegrist, 1977*). The feature selection methods were applied in a 5x20 nested cross-validation scenario provided with the 'RepeatedStratifiedKFold' method from the 'sklearn.model_selection' module of 'scikit-learn', setting the parameters 'n_splits'=5 and 'n_repeats'=20. From each cross-validation run, the selected variables were retained. The final feature sets for each selection method were determined by applying cABC analysis to the number by which each variable was selected in the set of 100 runs, from which the variables assigned to ABC category 'A' were then retained. The calculations were done using a Python implementation of above-mentioned R library, available at https://pypi.org/project/cABCanalysis/ (full method publication pending). The tuning of the algorithms led to the selection of LASSO and Ridge regression as regularization methods for SVM and logistic regression, respectively. Other hyperparameter settings include d=1200 trees with a maximum depth of 10 decisions for random forests trained to detect the time of the sample, 200 trees when trained to distinguish post-therapy samples from patients with neuropathy, a sigmoid kernel for SVM, or the selection of the 'newton-cg' solver for logistic regression.

A reduced feature set was determined from the sum count at which the variables had been selected in the different approaches. The sum score across the selections was subjected to cABC analysis, from which category 'A' provided a 'reduced' set of lipid mediators. To further narrow the focus to the most relevant variables, a nested cABC analysis was performed, which consists of repeating the cABC analysis for the items in category 'A' of a previous cABC analysis, providing a 'sparse' set of lipid mediators (separate method publication pending).

## Feature validation

Finally, it was assessed whether these sets of variables provided sufficient information for class separation in a sample not available during feature selection. The included algorithms were therefore trained with the full and reduced feature sets in a 5x20 nested cross-validation, using randomly selected subsets of 80% of the original training dataset for algorithm training, and applying the trained algorithms to random subsets comprising 80% of the validation dataset separated from the full original dataset before feature selection and classifier tuning. The balanced accuracy was used as the main parameter to evaluate the classification performance (*Brodersen et al., 2010*). In addition, the area under the receiver operator curve (roc-auc *Peterson et al., 1954*) was calculated. As elaborated above, the main goal was that the classification accuracy was better than by guessing, including that the 95% confidence interval (CI) of the balanced accuracy did not include the chance level of 0.5 (=50 %). The final set of lipid mediators was then analyzed for statistical differences according to the prior classification, that is pretreatment versus posttreatment and in the posttreatment sample, neuropathy versus no neuropathy, using Kruskal-Wallis tests (*Kruskal and Wallis, 1952*). Alpha correction for multiple testing was applied as suggested by *Bonferroni, 1936*.

## Lipid mediators informative for assigning samples to before or after therapy in an independent second patient cohort

The algorithms trained on the analysis cohort data were then applied to data from the independent second cohort without further modification or training. Specifically, after excluding variables or cases with >20% missing values and imputing the remaining missing values as described for cohort 1, analyzed data of cohort consisted of a matrix of size 52×239 (52 instances/samples and 239 different lipid mediators). The classifiers trained at the analysis cohort separated samples from the first from those from the second day in the second patient cohort at balanced accuracy of 0.6–0-.62 and roc-auc of 0.65–0.69 (Table 3). Using permuted data for training resulted in algorithms incapable to separate sample 1 from sample 2 better than by guessing. After analysis, SA1P differed between samples 1 and 2, which was consistent with its top ranking in the respective feature selection analysis. From the second patient cohort, six patients were excluded from the analysis due to more than 20% incomplete lipidomics data.

## Calcium Imaging with primary sensory neurons

Primary sensory neurons were cultured as described previously (*Sisignano et al., 2012*). For calcium imaging experiments, neurons were stained with Fura-2-AM (Thermo Fisher) for at least 60 min at 37 °C and washed afterwards twice with Ringer's solution consisting of 145 mM NaCl, 1.25 mM $CaCl_2 \times 2H_2O$, 1 mM $MgCl_2$ x6 $H_2O$, 5 mM KCl, 10 mM D-glucose, and 10 mM HEPES adjusted to a pH of 7.3. To investigate the effect of SA1P or LPC 24:0 on different TRP channels, sensory neurons were incubated with the lipids for 1 min at a concentration of 1 or 10 µM, respectively. The gold standard agonists for TRPV1 and TRPA1 were capsaicin (200 nM, 20 s) and AITC (allyl isothiocyanate, 75 µM, 30 s). Fingolimod was used at a concentration of 1 µM and pre-incubated for 1 hr prior to measurement. As a positive control, final stimulation with KCl (50 mM, 1 min) was used to depolarize all neurons. All stimulating compounds were dissolved in Ringer's solution to their final concentrations.

The calcium imaging data were analyzed using descriptive statistics. All calcium imaging data are presented as the mean ± SEM. Normal distribution was confirmed using the Shapiro-Wilk test. For experiments comparing only two groups, unpaired and heteroscedastic Student's t-tests were conducted following Welch's correction. When comparing more than two groups, one-way analysis of variance (ANOVA) was used, and for the comparison of more than three groups, two-way ANOVA was conducted. For all statistical analyses of the calcium imaging data, the software GraphPad Prism was used (version 9.5, GraphPad Software, Boston, MA, USA). Statistical significance was set at p-value <0.05.

## Results

All patients received paclitaxel as adjuvant or neoadjuvant therapy, without any other potentially neurotoxic substances. Of the 60 patients from our analysis cohort, two were excluded due to rescheduling of paclitaxel therapy. Blood samples were obtained from 31 patients before and after chemotherapy.

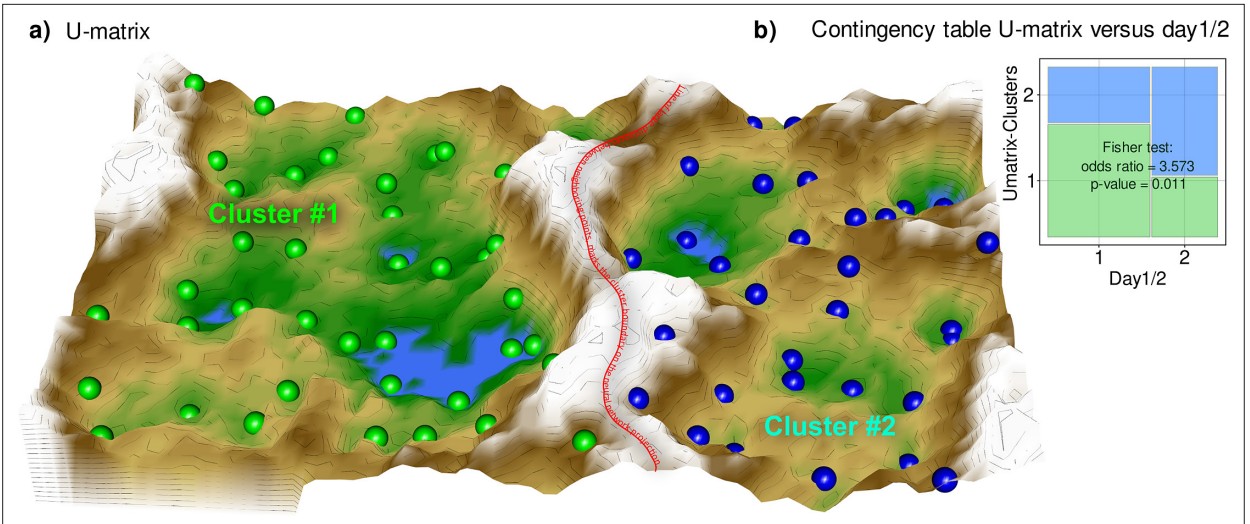

**Figure 2.** Results of a projection of the z-standardized log-transformed lipidomics data onto a lower-dimensional space by means of a self-organizing map of artificial neurons (bottom). (**a**) 3D display of an emergent self-organizing map (ESOM), providing a three-dimensional U-matrix visualization (*Thrun et al., 2016*) of distance-based structures of the serum concentrations of d=255 lipid mediators following projection of the data points onto a toroid grid of 4000 neurons where opposite edges are connected. The dots represent the so-called 'best matching units' (BMU), that is neurons on the grid that after ESOM learning carried a data vector that was most similar to a data vector of a sample in the data set. Only those neurons of the originally 4000 neurons are shown that carried vectors of cases from the present data set. Please also note that one BMU can carry vectors of several cases, that is the number of BMUs is not necessarily equal to the number of cases. A cluster structure emerges from visualization of the distances between neurons in the high-dimensional space by means of a U-matrix (*Izenmann, 2009*). The U-matrix was colored as a geographical map with brown or snow-covered heights and green valleys with blue lakes, symbolizing high or low distances, respectively, between neurons in the high-dimensional space. Thus, valleys left and right of the 'mountain range' in the middle indicate clusters and watersheds, that is the line of large distances between neighboring points, indicate borderlines between different clusters. Tat is, the mountain range with 'snow-covered' heights separates main clusters according to probe acquisition at day 1 or day 2, that is before and after treatment with paclitaxel. BMUs belonging to the two different clusters are colored green or bluish. (**b**) Mosaic plot of the prior classes (day 1 or day 2) versus the ESOM/Umatrix based clusters. The separation corresponded to the previous classification into pre- and post-therapy probes (day1/2). Cluster #1 was composed of more probes taken on day #1, while probes from day 2 were overrepresented in cluster #2. The figure has been created using the R software package (version 4.1.2 for Linux; https://CRAN.R-project.org/ *R Development Core Team, 2008*), R library 'ggplot2' (https://cran.r-project.org/package=ggplot2 (94)) and our R package 'Umatrix' (https://cran.r-project.org/package=Umatrix *Lötsch et al., 2018a*).

The online version of this article includes the following figure supplement(s) for figure 2:

**Figure supplement 1.** Results of a projection of the z-standardized log-transformed lipidomics data onto a lower-dimensional space by means of PCA.

Twenty lipid marker variables had >20% missing values. Following exclusion of these patients and variables and imputation, a data matrix for further analyses was obtained, sized 79×255 (79 data set instances, samples, and 255 different lipid mediators *Figure 1*). These included 48 samples drawn on day 1 before therapy and 31 samples drawn on day 2 after 12 cycles of paclitaxel therapy. On day 2, n=17 of the 31 patients had symptoms of neuropathy (54.9%), which is in line with previous clinical reports on occurrence and severity of paclitaxel-induced neuropathy (*Pachman et al., 2011*; *Scripture et al., 2006*). Most patients reported grade 1 neuropathy, although two patients experienced grade 3 neuropathy. Occurrence and degree of neuropathy were monitored 4.5 years after finishing chemotherapy. The neuropathy lasted for several months or, in many cases, still persisted for 4.5 years after chemotherapy at the last examination (*Supplementary file 1*).

## Results of unsupervised analysis of structure in the lipidomics data supporting prior knowledge

PCA yielded d=28 components with eigenvalues >1, which together explained 93.93% of the total variance in the lipid mediators (*Figure 2—figure supplement 1*). The d=86 lipid mediators that contributed most to the relevant PCs were identified based on the membership to category 'A' in the cABC analysis of the weighted variable contributions to each PC (*Figure 2—figure supplement 1*). This was carried over as one of the several feature-importance measures to the supervised analyses reported in the next chapter. On the emergent self-organizing map (ESOM, *Figure 2a*), a clear

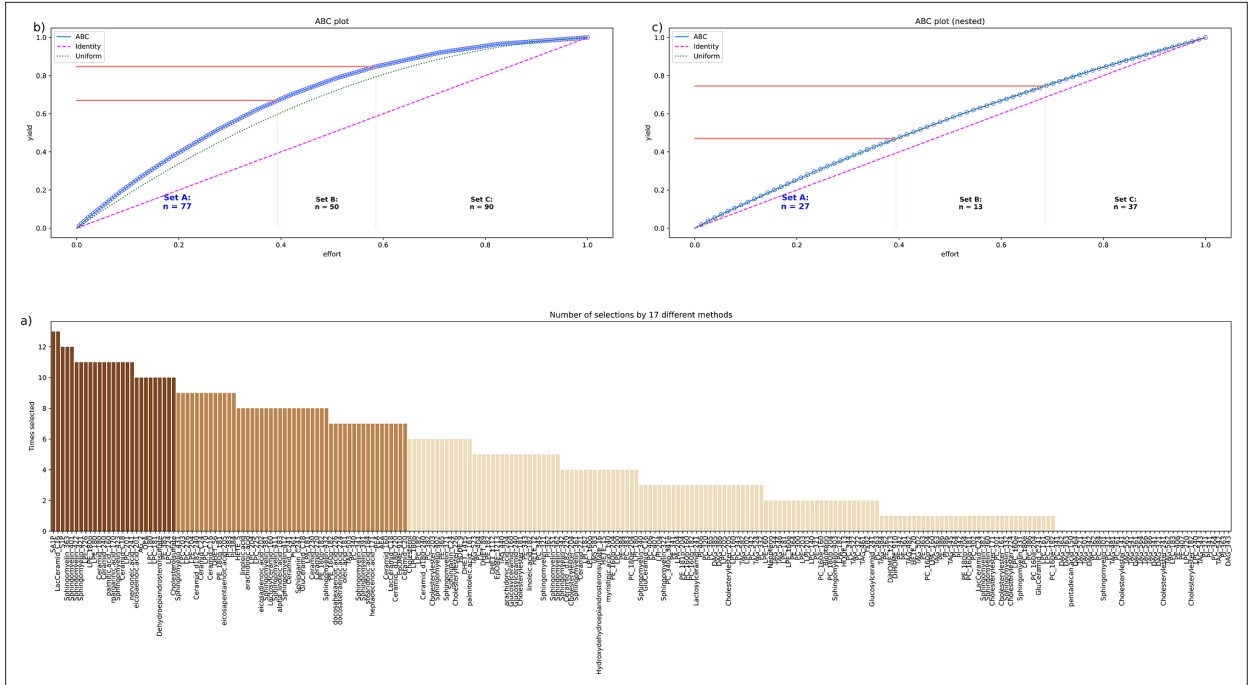

**Figure 3.** Identification of the lipid mediators that were most informative in assigning a sample to the pre- or post-therapy time point. Feature selection by 13 different methods. (**a**) Bar plot of the sum score of the number of selections for each lipid marker across the 13 methods. The colors of the bars correspond to the assignments of the lipid marker to category 'A' in a first cABC analysis (medium brown bars) and again to category 'A' in a second cABC analysis performed as a nested cABC analysis (dark brown bars). Light brown bars indicate that the marker was selected by too few methods to be considered further. (**b**) ABC plot (blue line) showing the cumulative distribution function of the sums of selections of each marker. The red lines show the boundaries between the CABC subsets 'A', 'B', and 'C'. Category 'A' with d=77 lipid mediators is considered to include the most relevant variables for sample time discrimination. (**c**) ABC plot of a nested cABC analysis performed on the d=77 mediators placed in category 'A' by a first cABC analysis. The figure was created using Python version 3.8.12 for Linux (https://www.python.org) with the seaborn statistical data visualization package (https://seaborn.pydata.org, *Waskom, 2021*) and our Python package 'ABCanalysis' (https://pypi.org/project/cABCanalysis/).

separation of two clusters was observed, which provided support that the lipid mediators contained a data structure contingent with the prior classification into pre- and posttherapy samples (Fisher's exact test: p=0.01054, odds ratio: 3.57 with 95% confidence interval 1.28–10.52; *Figure 2b*). The separation of samples on the ESOM also corresponded, but to a lesser extent, with the occurrence of neuropathy observed at the time of the 2nd blood sample (p=0.0328, odds ratio 0.16, 95% confidence interval 0.0137–1.022).

The results of the unsupervised analysis thus supported that the lipidomics data contained a structure contingent on a known prior classification. This supported the continuation of data analysis with supervised methods to determine which of the lipids carried relevant information to assign a probe to a particular prior class.

## Results of supervised analyses identifying lipid mediators relevant to the class structure

### Lipid mediators informative for assigning samples to before or after paclitaxel therapy

Based on the majority vote of the different approaches to feature selection including PCA importance and further univariate and multivariate feature selection methods, d=77 lipid mediators were found to provide relevant information on whether a sample was collected before or after paclitaxel therapy (*Figure 3*). When statistical (logistic regression) and machine learning (random forests, support vector machines - SVM) algorithms were trained with this set of lipid mediators, the assignment of a sample to day 1 or 2 was well above the guessing level (*Table 1*). By contrast, when the training data were randomly permuted, the performance fell to a balanced accuracy of 0.5, that is guessing level, which established that the obtained class assignment in the non-permuted scenario had not been due to

**Table 1.** Internal validation of the sets of lipid mediators resulting from the feature selection analysis. The different classifiers (linear support vector machine, SVM, random forests, and logistic regression) were trained with subsets of the training data set with all variables d=255 lipid mediators as 'full' feature set and with the d=77 or d=27 lipid mediators that had resulted from the recursive cABC analysis applied on the sum score of selections by 17 different feature selection methods as 'reduced' or 'sparse' feature sets, respectively. The trained classifiers were applied to a validation sample comprising 20% of the data that had been removed in a class-proportional manner from the dataset at the beginning of feature selection and had not been touched until used in the classifier validation task presented in this table. In addition, the validation task was repeated with training the classifiers with permuted lipid mediators to observe possible overfitting. Shown are the medians and nonparametric 95% confidence intervals (2.5th to 97.5th percentiles) from 5x20 nested cross-validation runs.

| Classifier | Performance measure | Feature set | | | |
|---|---|---|---|---|---|
| | | Full | Reduced | Reduced permuted | Sparse |
| | Number of lipid mediators | 255 | 77 | 77 | 27 |
| SVM | Balanced accuracy | 0.7 (0.48–0.92) | 0.78 (0.61–1) | 0.48 (0.23–0.76) | 0.75 (0.54–0.91) |
| Random forests | | 0.7 (0.56–0.83) | 0.75 (0.58–0.85) | 0.46 (0.24–0.75) | 0.74 (0.55–0.88) |
| Logistic regression | | 0.7 (0.52–0.85) | 0.77 (0.58–0.92) | 0.48 (0.29–0.7) | 0.7 (0.49–0.89) |
| SVM | roc-auc | 0.88 (0.67–1) | 0.95 (0.85–1) | 0.48 (0.16–0.77) | 0.9 (0.81–0.99) |
| Random forests | | 0.86 (0.76–0.95) | 0.88 (0.81–0.98) | 0.48 (0.21–0.81) | 0.9 (0.8–1) |
| Logistic regression | | 0.81 (0.64–1) | 0.88 (0.75–1) | 0.46 (0.16–0.79) | 0.86 (0.69–0.98) |

overfitting. In addition, by rerunning the cABC analysis on the mediators assigned to subset 'A' in the first run ('recursive' cABC analysis *Lötsch and Ultsch, 2023b*), the informative set of lipid mediators could be further reduced to d=27 (*Table 2*, *Table 3*). With these mediators, SVM and random forest were still able to detect whether an instance of a lipidomics dataset was from before or after paclitaxel treatment at a balanced accuracy better than expected from guessing. In summary, with the top hits ('sparse' feature set), three different algorithms (logistic regression, random forests, support vector machines) could be trained to identify, in new cases, whether a blood sample was drawn before or after paclitaxel therapy with a median balanced accuracy of up to 90%.

## Lipid mediators informative for assigning post-paclitaxel therapy samples to neuropathy

The n=31 samples from day 2 were probably too small to detect whether a sample was from a patient with neuropathy. Although the three algorithms detected neuropathy in new cases, unseen during training, at balanced accuracy of up to 0.75, while only the guess level of 0.5 was achieved when using permuted data for training, the 95% CI of the performance measures was not separated from guess level. Therefore, multivariate feature selection was not considered a valid approach, since it requires that the algorithms from which the feature importance is read can successful perform their task of class assignment (*Lötsch and Ultsch, 2023a*). Therefore, univariate methods (Cohen's d, FPR, FWE) were preferred, as well as a direct hypothesis transfer of the top hits from the above-mentioned day1/2 assessments to neuropathy. Classical statistics consisting of direct group comparisons using Kruskal-Wallis tests (*Kruskal and Wallis, 1952*) were performed. The small set of d=3 lipid mediators that emerged from all three univariate methods as top hits for neuropathy included sphingolipid sphinganine-1-phosphate (SA1P), also known as dihydrosphingosine-1-phosphate

**Table 2.** Lists of lipid mediators that were most informative in assigning a sample (i) to the first or second sampling time point or (ii) a sample from the second time point to a patient with or without neuropathy.

Abbreviations: SA1P: sphinganine-1-phosphate, S1P: sphingosine-1-phosphate, LPE: lysophosphatidylethanolamine, LPC: lysophosphatidylcholine, 2-AG: 2-arachidonoylglycerol, OEA. Oleoylethanolamide.

**Sample 1 versus sample 2**

| | | | | |
|---|---|---|---|---|
| SA1P | Sphingomyelin 42:1 | Palmitic acid 16:0 | Eicosaeinoic acid 20:1 | PE 38:5 |
| LacCeramid C16 | LPE 22:6 | Margaritic acid 17:0 | 2-AG | LPC 22:4 |
| S1P | LPE 18:0 p | Sphingomyelin 42:3 | OEA | Cholesterolsulfate |
| Sphingomyelin 36:3 | LPE 18:0 | LPC 18:0 | Sphingomyelin 40:1 | |
| Ceramide 18:0 | LPC 20:1 | LPC 18:1 | Sphingomyelin 42:2 | |
| Ceramide 24:0 | Nervoneic acid 24:1 | Dehydroepiandrosterone sulfate | | |

**Sample 2: neuropathy versus no neuropathy**

| | | |
|---|---|---|
| SA1P | Sphingomyelin 33:1 | Sphingomyelin 43:1 |

(DH-S1P) sphingomyelin 33:1, and sphingomyelin 43:1. Statistical group comparisons verified that the three mediators differed significantly between samples from neuropathy-positive and neuropathy-negative patients (*Figure 4*). An overview of the distribution of the entire lipid marker data is shown in *Figure 4—figure supplement 1*.

## Biological in-vitro validation of the machine learning-based results

The results of the supervised analysis thus established a limited set of lipid mediators to be regulated in association with paclitaxel therapy or with its side effect of inducing neuropathy. The top hit was sphinganine-1-phosphate (SA1P), providing a basis for in vitro validation of its biological effects in the present context of neuropathy.

**Table 3.** External validation of the classifiers in an independent patient cohort.

The different classifiers (linear support vector machine, SVM, random forests, and logistic regression) were trained with subsets of the training data set from the analysis cohort using the 'sparse' feature set. The trained classifiers were then applied to an independent second patient cohort. In addition, the validation task was repeated with permuted information from lipid mediators to observe possible overfitting. Shown are the medians and nonparametric 95% confidence intervals (2.5th to 97.5th percentiles) from 5x20 nested cross-validation runs.

| Classifier | Performance measure | Sparse | Sparse permuted |
|---|---|---|---|
| | Number of lipid mediators | 27 | 27 |
| SVM | Balanced accuracy | 0.6 (0.52–0.68) | 0.5 (0.49–0.51) |
| Random forests | | 0.62 (0.5–0.68) | 0.52 (0.37–0.65) |
| Logistic regression | | 0.62 (0.54–0.74) | 0.51 (0.31–0.69) |
| SVM | roc-auc | 0.65 (0.54–0.73) | 0.51 (0.19–0.72) |
| Random forests | | 0.66 (0.55–0.75) | 0.52 (0.24–0.76) |
| Logistic regression | | 0.69 (0.6–0.78) | 0.49 (0.25–0.78) |

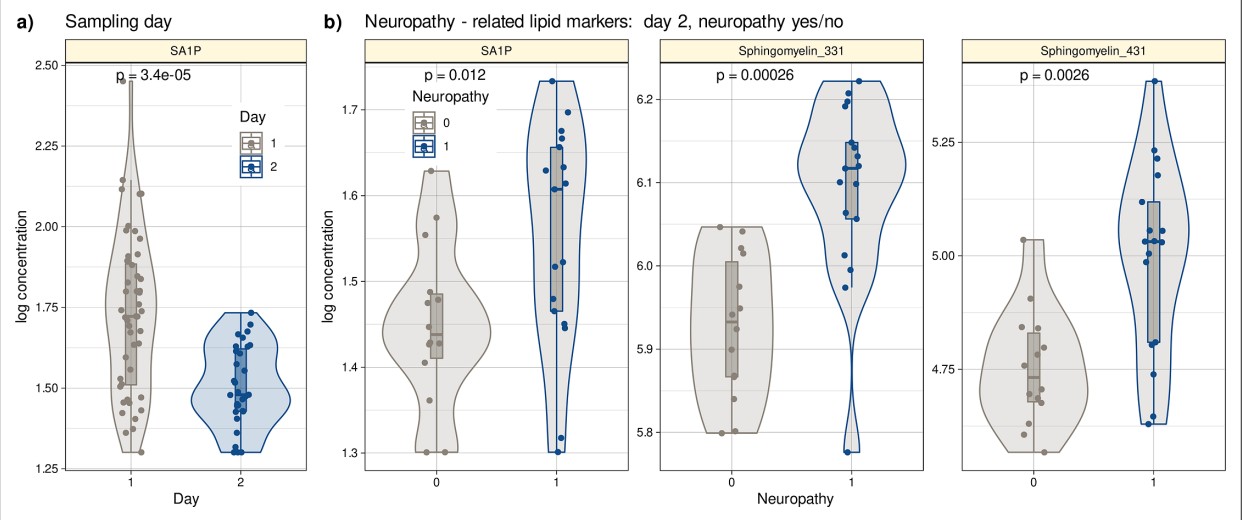

**Figure 4.** Log₁₀-transformed concentrations of lipid mediators shown to be informative for assigning a post-therapy sample to a patient with neuropathy or a patient without neuropathy. Individual data points are presented as dots on violin plots showing the probability density distribution of the variables, overlaid with box plots where the boxes were constructed using the minimum, quartiles, median (solid line inside the box) and maximum of these values. The whiskers add 1.5 times the interquartile range (IQR) to the 75th percentile or subtract 1.5 times the IQR from the 25th percentile. (**a**) Concentrations of SA1P (top hit for sample 1 versus sample 2 segregation) are presented separately for the first and second samples. (**b**) Concentrations of the top lipid mediators for neuropathy versus no neuropathy in the second sample presented separately for neuropathy-positive and -negative samples. The results of the group comparison statistics (Kruskal-Wallis tests **Kruskal and Wallis, 1952**) are given at the top of the graphs. The figure has been created using the R software package (version 4.1.2 for Linux; http://CRAN.R-project.org/, **R Development Core Team, 2008**) and the R library 'ggplot2' (https://cran.r-project.org/package=ggplot2, **Wickham, 2009**).

The online version of this article includes the following figure supplement(s) for figure 4:

**Figure supplement 1.** Plasma lipids from the patient cohort.

Calcium imaging measurements were performed on primary sensory neurons obtained from the murine dorsal root ganglia from at least four different animals. We stimulated the neurons with 1 and 10 µM SA1P (ranked as the primary hit) or LPC 24:0 (ranked as one of the least relevant lipids by machine learning analysis). We observed that SA1P caused a direct calcium transient in approximately 11.7% of KCl-responsive sensory neurons (**Figure 5a and b**). However, LPC 24:0 did not induce any notable activation of sensory neurons at concentration of 1 and 10 µM (**Figure 5—figure supplement 1**). To further characterize the SA1P-responding neurons, we investigated their responsiveness to agonists of the TRP channels TRPV1 (capsaicin) and TRPA1 (AITC, allyl isothiocyanate), both of which are hallmarks of subpopulations of primary sensory neurons (**Julius, 2013**). Stimulating SA1P-responsive neurons with capsaicin and AITC revealed that 73% of these neurons also responded to capsaicin and 25% of them responded to AITC, whereas only 9.6% responded to both stimuli. Neurons were identified as responders to KCl (50 mM, 1 min; **Figure 5c and d**).

To identify the receptors or channels responsible for SA1P-mediated calcium transients in sensory neurons, the selective TRPV1 antagonist AMG9810 was used. Neurons were stimulated twice with SA1P (1 µM, 1 min) and AMG9810 (1 µM or vehicle) was added two minutes prior to the second SA1P stimulus. The second SA1P response was entirely abolished when the neurons were treated with AMG9810, but not with the vehicle (**Figure 6a–c**). The potency of AMG9810 was validated using the same measurement protocol as before but with capsaicin (200 nM, 20 s) instead of SA1P, which is the gold standard agonist of TRPV1. AMG9810 completely blocked the second capsaicin response (**Figure 6d**). The involvement of S1P-receptors previously suggested to be the receptors for SA1P (**Magaye et al., 2019**), was evident by studying S1PR1 and S1PR3 as the most highly expressed S1P receptors in sensory neurons (**Quarta et al., 2017**), which are also targets of the approved drug fingolimod. Sensory neurons were incubated with fingolimod or vehicle for one hour and stimulated the neurons with SA1P (**Figure 6e and f**). Comparing fingolimod- and vehicle-treated neurons, we observed that the response intensity to SA1P was similar (**Figure 6g**), while the number of neurons responding to SA1P was significantly decreased after fingolimod treatment (**Figure 6h**).

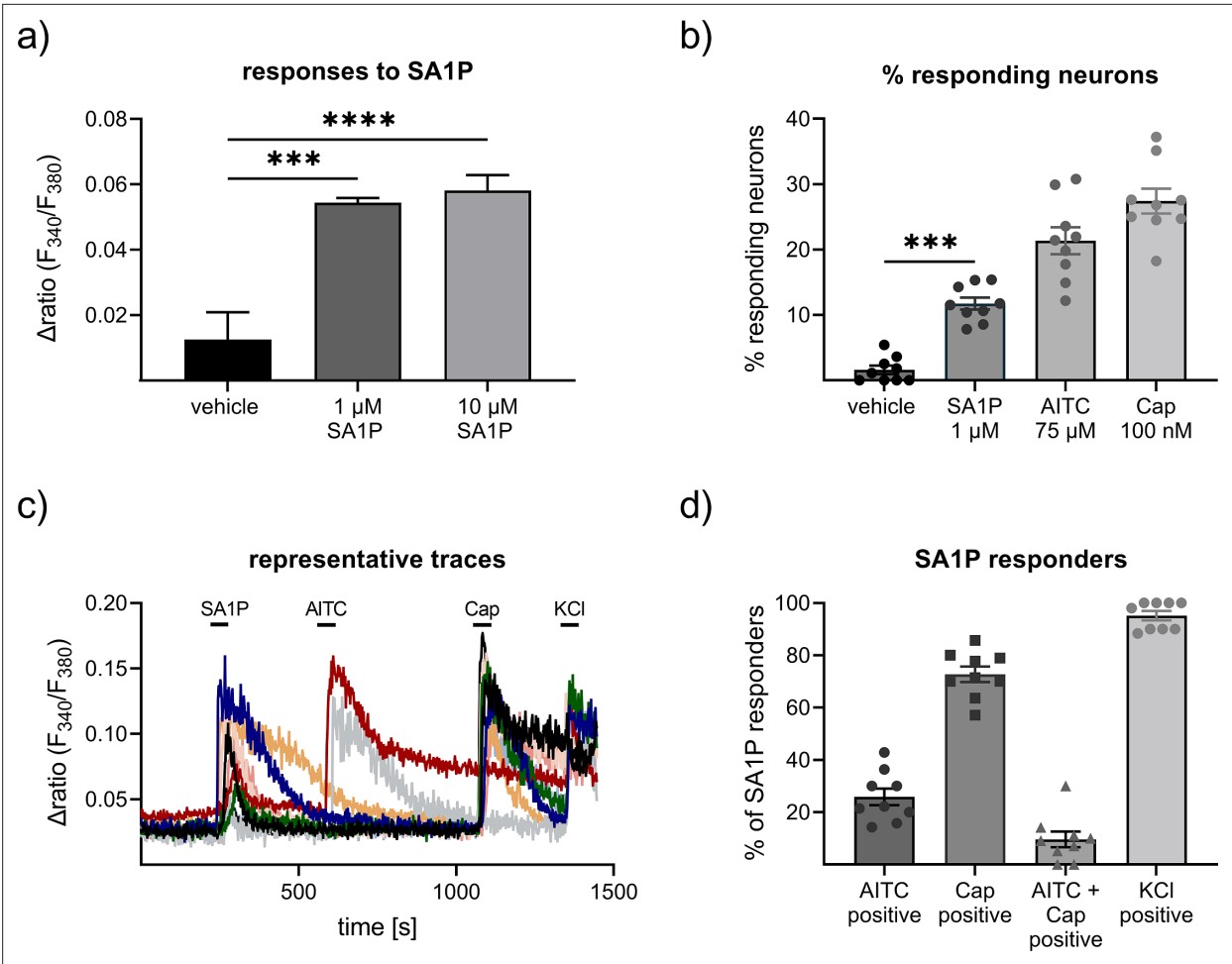

**Figure 5.** Effects of sphinganine-1-phosphate (SA1P) on primary sensory neurons. (**a**) Neurons were stimulated with SA1P (1 or 10 µM, 1 min or vehicle (0.7% methanol (v/v)). (**b**) Percentage of responding neurons to vehicle 0.7% methanol (v/v), 1 min), SA1P (1 µM, 1 min), AITC (allyl isothioncyanate, 75 µM, 30 s), or capsaicin (Cap, 200 nM, 20 s). (**c**) Representative traces of SA1P-responding neurons and their response to AITC, capsaicin and KCl. (**d**) Percentage of SA1P-responding neurons responding to AITC, capsaicin (Cap), AITC and capsaicin and KCl. Data are shown as mean ± SEM from at least six measurements per condition with at least 40 neurons per measurement, * p<0.05, ** p<0.01, *** p<0.01, One-way ANOVA.

The online version of this article includes the following figure supplement(s) for figure 5:

**Figure supplement 1.** Effects of lysophosphatidylcholine 24:0 (LPC 24:0) on primary sensory neurons.

## Support of the main results in an independent second patient cohort

The study lacked a separate validation cohort with similar data to the main cohort. However, an independent second cohort was available from another hospital (Oncological Center in Offenbach, Germany). This cohort consisted of 28 patients treated with the 'paclitaxel weekly' regimen (paclitaxel 80 mg/m² , once weekly) as adjuvant or neoadjuvant therapy for breast or ovarian cancer. All patients provided informed consent into study participation and publication of the results. Blood samples were available from routine collections and were analyzed by LC-MS/MS (*Figure 7—figure supplement 1*, *Supplementary file 3*). In contrast to the main cohort, plasma from patients in the second cohort was collected after six cycles of paclitaxel treatment due to local routines. Therefore, the second cohort cannot be considered a state-of-the-art validation cohort. In addition, only six patients in this cohort had neuropathy after chemotherapy (26.5%), all with grade 1 neuropathy (*Supplementary file 3*). Despite these limitations, algorithms trained with lipid information from cohort 1 were able to successfully identify whether a probe was taken before or after paclitaxel therapy in the second cohort at a better than guessing level. Furthermore, a trend toward different SA1P concentrations in the plasma of patients after paclitaxel treatment was observed (p=0.086, *Figure 7*).

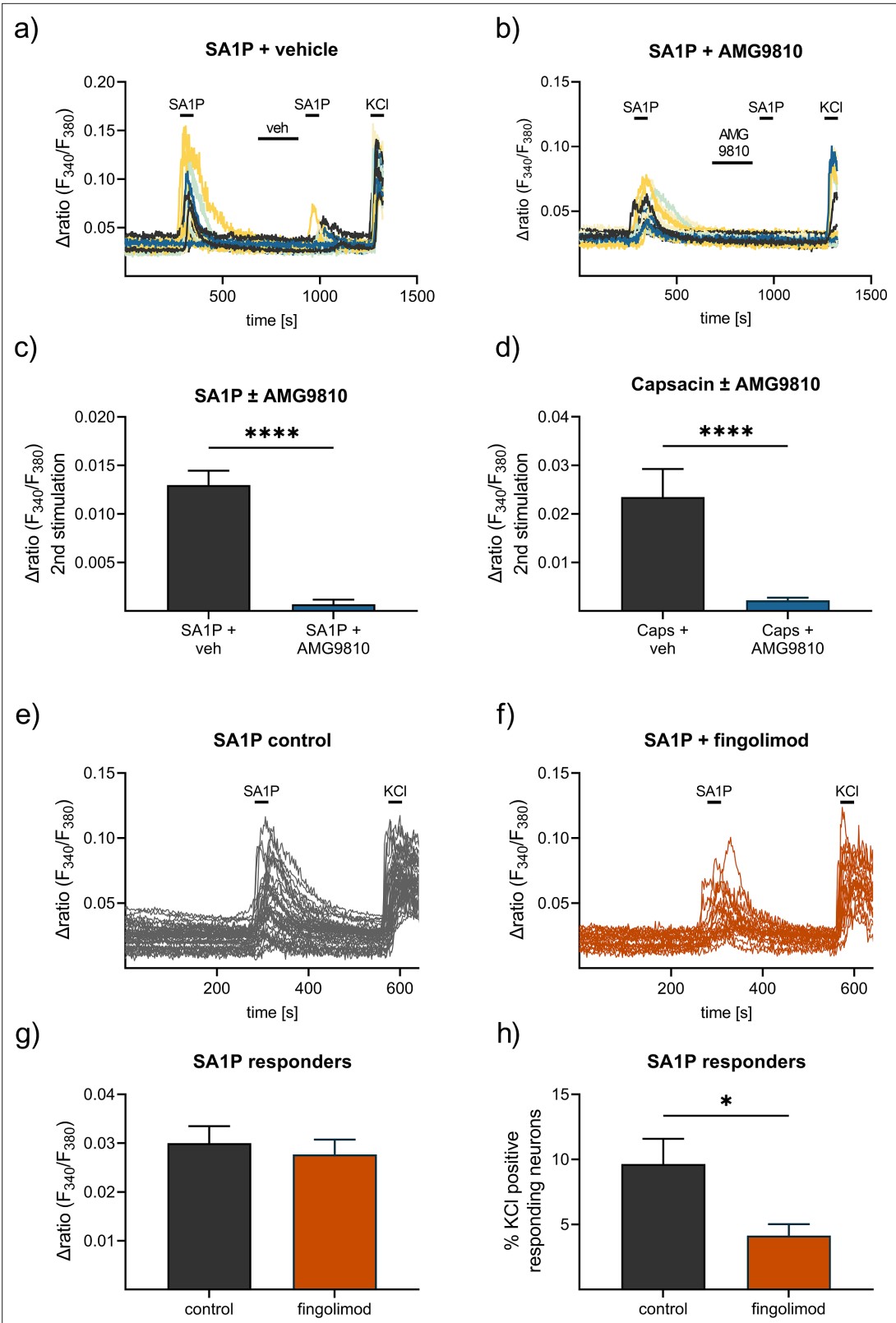

**Figure 6.** Contribution of TRPV1 and S1P receptors to SA1P-mediated calcium-influx in sensory neurons. Sensory neurons were stimulated with SA1P twice (1 µM, 1 min) and either (**a**) vehicle (DMSO 0.003% (v/v), 2 min) or (**b**) the TRPV1 antagonist AMG9810 (1 µM, 2 min) prior to the second SA1P stimulus. Cells were depolarized with KCl (50 mM, 1 min) at the end of each experiment. (**c**) Statistical analysis of the amplitude of SA1P-mediated calcium transients in sensory neurons treated with either vehicle or AMG9810 (blue). (**d**) Statistical analysis of the amplitude of capsaicin-mediated

*Figure 6 continued on next page*

*Figure 6 continued*

calcium transients (Caps, 100 nM, 20 s) in sensory neurons treated with either vehicle or AMG9810 (blue). (**e**, **f**) Sensory neurons were stimulated with SA1P after preincubation with the S1P1 receptor modulator fingolimod (1 µM, 1 hr) or control. (**g**) Statistical analysis of the amplitude of SA1P-mediated calcium transients (1 µM, 1 min) in sensory neurons treated with either vehicle or fingolimod (1 µM, 1 hr, orange). (**h**) Statistical analysis of the number of SA1P-responding neurons (as % of KCl-positives) after treatment with either vehicle or fingolimod (1 µM, 1 hr, orange). Data represents mean ± SEM from at least five measurements per condition with at least 25 neurons per measurement, * p<0.05, ** p<0.01, *** p<0.01, Student's t-test with Welch's correction.

## Discussion

For paclitaxel, therapeutic doses range from 80 to 225 mg/m². As CIPN symptoms are dose-dependent, the number of PIPN patients that receive a high paclitaxel dose is higher than the number of PIPN patients receiving a low dose. The cumulative threshold dose above which paclitaxel causes sensory neuropathy is 300 mg/m² (*Park et al., 2013*). In our study, we mainly used a low dose

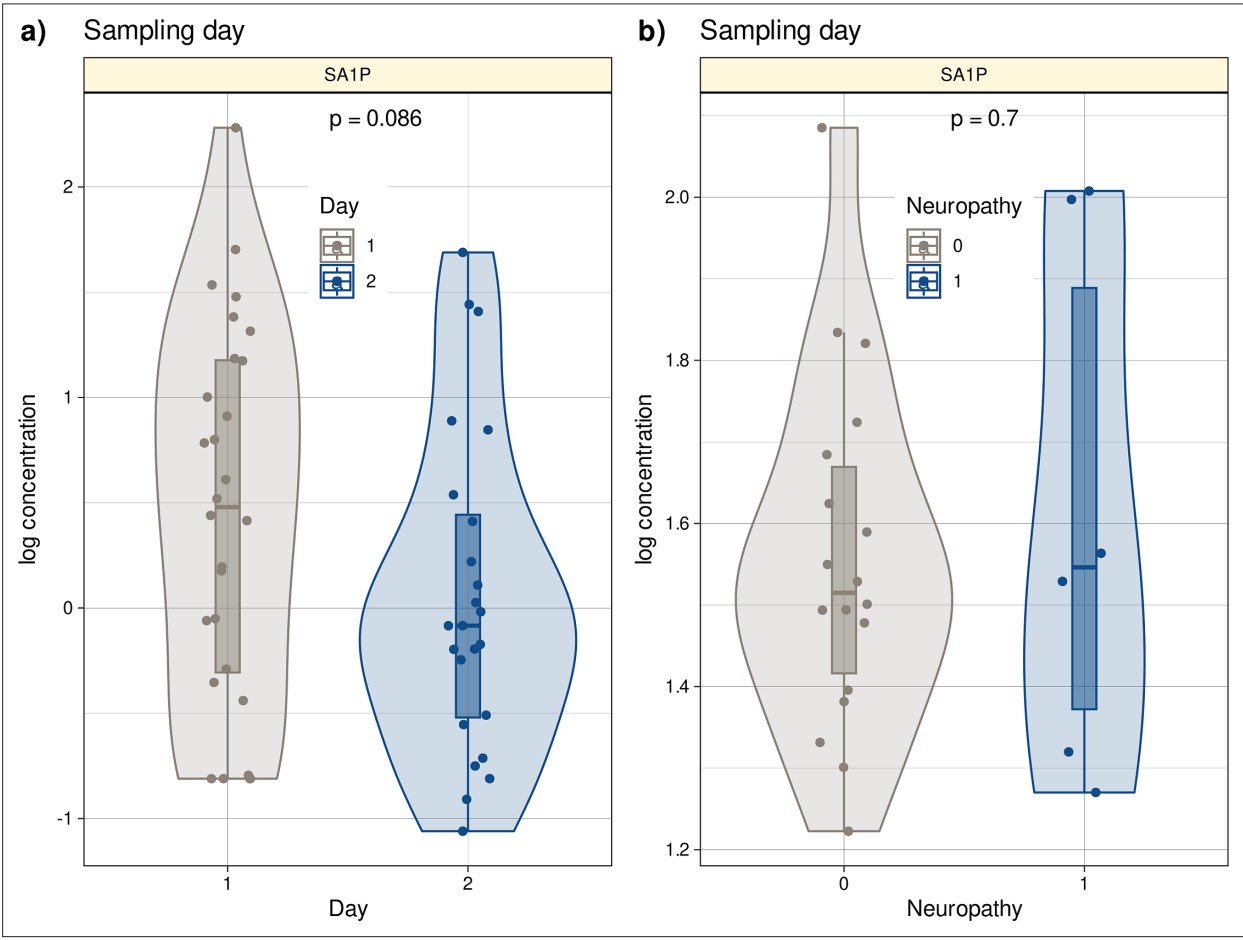

**Figure 7.** Log10-transformed concentrations of SA1P in the second patient cohort. Individual data points are presented as dots on violin plots showing the probability density distribution of the variables, overlaid with box plots where the boxes were constructed using the minimum, quartiles, median (solid line inside the box) and maximum of these values. The whiskers add 1.5 times the interquartile range (IQR) to the 75th percentile or subtract 1.5 times the IQR from the 25th percentile. (**a**) Concentrations of SA1P (top hit for sample 1 versus sample 2 segregation) are presented separately for the first and second samples. (**b**) Concentrations of S1AP in the second sample are shown separately for neuropathy-positive and -negative samples. Day 1 represents the timepoint before starting chemotherapy. Day 2 represents the timepoint after 12 cycles of paclitaxel chemotherapy. The results of the t-test group comparison statistics are given at the top of the graphs. The figure has been created using the R software package (version 4.1.2 for Linux; http://CRAN.R-project.org/, *R Development Core Team, 2008*) and the R library 'ggplot2' (https://cran.r-project.org/package=ggplot2, *Wickham, 2009*).

The online version of this article includes the following figure supplement(s) for figure 7:

**Figure supplement 1.** Plasma lipids from the independent second patient cohort.

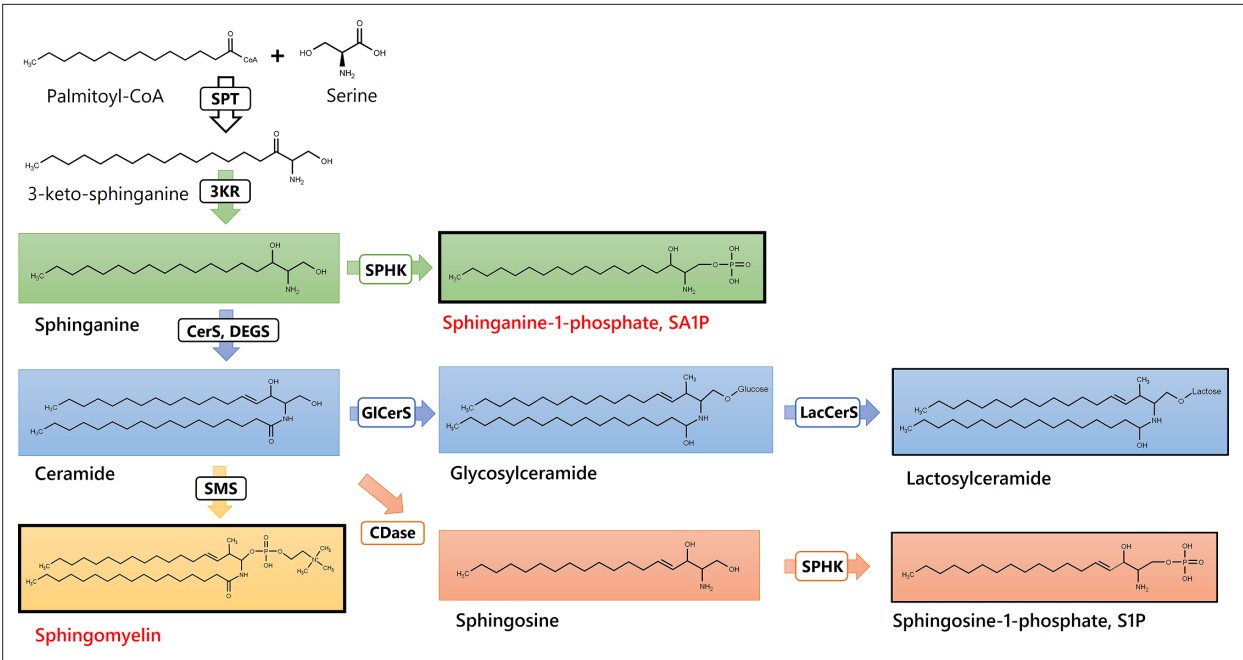

**Figure 8.** Sphingolipids and Ceramides (SPT: Serine palmitoyl-transferase; 3KR: 3-ketosphinganine reductase; SPHK: Sphingosine kinase; CerS: Ceramide synthase; DEGS: Dihydroceramide desaturase, GlCerS: Glucosylceramide synthase; LacCerS: Lactosylceramide synthase; SMS: Sphingomyelin synthase; CDase: Ceramidase). Structures were drawn with ChemDraw 20.

paclitaxel, because this therapeutic regimen is the most widely used paclitaxel monotherapy. Previous studies report an occurrence of neuropathy with this therapeutic regimen is around 50–70%, and most patients (80–90%) are expected to experience Grade 1 neuropathy after 12 weeks (**Gornstein and Schwarz, 2014**; **Hershman et al., 2014**; **Yang and Horwitz, 2017**). Our results are within the range reported by these previous studies.

SA1P induced a direct calcium transient in sensory neurons dependent on sphingosine 1-phosphate receptors (S1PR) and the transient receptor potential vanilloid 1 (TRPV1) channel. The results suggest that lipids are altered during paclitaxel treatment and that alterations in sphingolipid metabolism may be critical for the development of paclitaxel-induced peripheral neuropathy in patients. The final ('sparse') set of lipids regulated between sampling days (before and after paclitaxel treatment) was enriched for sphingolipids. Specifically, sphingolipids were significantly overrepresented among the top hits of lipids regulated between sampling days (Fisher's exact test: p=0.01), that is while 46 of the original 255 lipid mediators were sphingolipids (18%) (**Supplementary file 2**), 11 of the 27 members of the final sparse marker set (40.7%) belonged to the group of sphingolipids. The main pathway in which the top hits are involved is shown in **Figure 8**. The classification of patients with regard to neuropathy after paclitaxel treatment was reflected in lipidomics in another sphingolipid, that is dihydrosphingosine sphinganine-1-phosphate (SA1P), which was elevated in patients with neuropathy.

The results from an unbiased machine-learning-based analysis are in line with those of previous reports that had used classical statistics mainly. For example, S1P was elevated in the spinal cord of mice after bortezomib treatment and during bortezomib-induced neuropathic pain. Blocking S1P1 receptor S1P1R with fingolimod effectively reduced bortezomib-induced mechanical hypersensitivity in vivo (**Stockstill et al., 2018**). Interestingly, targeting the S1P-S1P1R-axis was also found to reduce paclitaxel-induced neuropathic pain in vivo in a preclinical study (**Janes et al., 2014**). Further agreements of the present results relate to previous preclinical reports highlighting the significance of the sphingolipid pathway in persistent and neuropathic pain states (**Stockstill et al., 2018**; **Squillace et al., 2020**; **Chen et al., 2019**). In addition, the S1P signaling axis was observed to be relevant in neuropathy and chemotherapy-induced neuropathic pain, which led to the suggestion of targeting S1P receptors as a novel approach to ameliorate chemotherapy-induced neuropathy and neuropathic pain (**Janes et al., 2014**; **Wang et al., 2020**; **Salvemini et al., 2013**; **Becker et al., 2020**; **Chua et al., 2020**). Taken together, present results from lipid screening and unbiased machine-learning approach

point towards a relevant contribution of sphingolipid signaling in paclitaxel-induced neuropathy in patients, mainly via sphinganine-1-pshophate and sphingomyelins 33:1 and 43:1.

Sphinganine is a key branching point in the sphingolipid pathway, where it can either be acylated to form dihydroceramides or phosphorylated to SA1P by sphingosine kinases (**Merrill, 2011**). Accumulation of sphinganine has been previously associated with reduced activity of ceramide synthase CerS2 (**Pewzner-Jung et al., 2010**). Interestingly, low CerS2 expression is a hallmark of various tumors (**Zhang et al., 2019**). It is conceivable that a subgroup of the patient cohort still exhibits low CerS2 expression after paclitaxel treatment, which is associated with higher SA1P levels and a higher occurrence of neuropathy. We also identified SA1P as a potential proalgesic lipid mediator, as it causes direct calcium transients in approximately 10% of sensory neurons. This effect is mediated, at least in part, by S1P receptors S1P1 and S1P3 and the TRPV1 channel. The TRPV1 channel has previously been identified as an important mediator of neuropathic pain and is associated with exacerbated activity of sensory neurons during paclitaxel-induced neuropathic pain (**Jardín et al., 2017**; **Hara et al., 2013**; **Kamata et al., 2020**). Our data indicate that an S1P receptor modulator, such as fingolimod may be a potential treatment strategy for reducing the proalgesic effect of SA1P and possibly for reducing paclitaxel-induced neuropathy in patients.

Several other lipids were found in the extended list of hits previously associated with chemotherapy-induced neuropathy or acute pain in preclinical studies, such as LPC 18:1, sphingosin-1-phosphate (S1P), and 9,10-EpOME. LPC 18:1 has previously been identified as an endogenous activator of TRPV1 and TRM8 and was found at elevated levels in murine DRGs 24 hr after oxaliplatin treatment. This may contribute to oxaliplatin-induced acute pain (**Rimola et al., 2020**). Similarly, 9,10-EpOME was elevated in the DRGs of paclitaxel-treated mice. locking its synthesis with the CYP2J2-inhibitor and the approved drug telmisartan was shown to reduce acute paclitaxel-induced mechanical hypersensitivity in vivo and prevent paclitaxel-induced mechanical allodynia by pretreatment (**Sisignano et al., 2016**). In addition, the direct TRPV1 agonist LPA 18:1 was found in the extended list of hits. Lipid was shown to bind to the C-terminal binding site of TRPV1 to increase the opening probability of the channel (**Nieto-Posadas et al., 2012**). Other signaling lipids with potential proalgesic effects and potential TRP channel activators that have been described previously have also been identified as crucial for group separation by our unbiased machine-learning approach, which strengthens the presumption that the identified lipids may indeed be connected with paclitaxel neurotoxicity and paclitaxel-induced neuropathy in patients. Additionally, we found several precursor fatty acids in the extended list of hits, including arachidonic acid, linoleic acid, and palmitic acid. These results imply a major dysregulation of lipids after paclitaxel treatment, leading to enhanced plasma levels of precursors for eicosanoids and oxidized linoleic acid metabolites, which may explain the observed enhanced concentrations of the eicosanoids of 5-HETE, 5,6-DHET, and the oxidized linoleic acid metabolites 9,10-EpOME, 9- and 13-HODE.

Several limitations need to be addressed. First, the size of our patient cohort (n=60 patients, n=31 patients who provided blood samples before and after paclitaxel chemotherapy) allowed for errors due to interindividual differences. Second, the paclitaxel treatment regimen differed in some patients included in this study. While the majority of the patients received paclitaxel as 'pacli weekly' regime consisting of a weekly dose of 80 mg/m² for 12 consecutive weeks in, some few patients received paclitaxel doses up to 220 mg/m² or a combination chemotherapy consisting of carboplatin/paclitaxel. These variabilities within the patient cohort hamper interindividual correlations between plasma lipid mediator concentrations and paclitaxel-induced neuropathy throughout the treatment course. Third, the assessment of peripheral neuropathy in patients was performed according to the guidelines of the NCI Common Terminology Criteria for Adverse Events (CTCAE) v5.0, which ranks the severity of neuropathy into five grades but is rather focused on general adverse events of chemotherapy rather than specifically assessing peripheral neuropathy in detail. We did not perform any neurological testing of sensory parameters, such as quantitative sensory testing (QST) (**Maier et al., 2010**), to determine the sensory status quo of the patients.

The identification of sphinganine-1-phosphate (SA1P) as a key lipid marker delineating the effects of paclitaxel was based on a comprehensive analysis performed by a 'mixture of experts' of machine learning and classical statistical methods, coupled with in vitro molecular experiments performed on sensory neurons. A rigorous validation process was applied to ensure that the results did not depend on the properties of any single statistical or AI model. This included verifying that the lipidomics

data intrinsically supported the existing class structure (days 1/2) using two different data projection methods (PCA, neural network). In addition, the identification of the most relevant lipids associated with past exposure to paclitaxel relied on univariate and multivariate feature selection, and the results were verified by testing three different algorithms that finally could successfully identify from the concentrations of the top hits whether a blood sample was drawn before or after paclitaxel therapy. In addition, all data analyses were subjected to rigorous cross-validation. Results were first validated on a sample split from cohort #1 prior to any feature selection and classifier training, and again on the data from cohort #2. Then, the leading hit from machine learning, SA1P, was validated as a highly active mediator in a neuronal cell model, while a lipid maker deemed irrelevant by machine learning actually showed ineffectiveness in the cell model. The mechanistical details of altered sphingolipid metabolism reflected in their plasma concentrations of paclitaxel patients need to be investigated by further studies.

## Conclusions

Here, we demonstrate that the combination of state-of-the-art lipidomics using LC-MS/MS, LC-QTOF-MS, and machine learning-based data analysis can robustly lead to the generation of test-able hypotheses and the identification of biologically relevant signaling mediators of neuropathy in an unbiased manner. Lipidomic profiles were compared within the same patients, allowing analysis of individual paclitaxel-induced lipidome changes in the same patients. These analyses led to the identification of a lipid mediator that can directly activate calcium transients in sensory neurons, thereby modulating nociceptive processing and sensory neuron activity. The identified SA1P, through its receptors, may provide a potential drug target for co-therapy with paclitaxel to reduce one of its major and therapy-limiting side effects.

## Acknowledgements

We thank Drs Tabea Osthues, Béla Zimmer and Vittoria Rimola, as well as Mr. Maksim Sendetski for their help in patient sample processing.

## Additional information

### Funding

| Funder | Grant reference number | Author |
|---|---|---|
| Deutsche Forschungsgemeinschaft | SFB1039 A09 | Saskia Wedel<br>Gerd Geisslinger<br>Marco Sisignano |
| Deutsche Forschungsgemeinschaft | SFB1039 Z01 | Lisa Hahnefeld<br>Carlo Angioni<br>Yannick Schreiber<br>Sandra Trautmann<br>Gerd Geisslinger |
| Fraunhofer-Gesellschaft | Foundation Project: Neuropathic Pain | Jörn Lötsch<br>Gerd Geisslinger<br>Marco Sisignano |
| Fraunhofer Cluster of Excellence Immune-Mediated Diseases | | Gerd Geisslinger<br>Marco Sisignano |
| Deutsche Forschungsgemeinschaft | DFG LO 612/16-1 | Jörn Lötsch |
| Fraunhofer-Gesellschaft | Leistungszentrum Innovative Therapeutics (TheraNova) | Gerd Geisslinger<br>Marco Sisignano |

The funders had no role in study design, data collection and interpretation, or the decision to submit the work for publication.

## Author contributions
Jörn Lötsch, Resources, Data curation, Software, Formal analysis, Validation, Visualization, Methodology, Writing – original draft, Writing – review and editing; Khayal Gasimli, Conceptualization, Supervision, Funding acquisition, Investigation, Writing – original draft, Writing – review and editing; Sebastian Malkusch, Data curation, Software, Formal analysis, Validation, Investigation, Methodology; Lisa Hahnefeld, Resources, Formal analysis, Validation, Investigation, Methodology; Carlo Angioni, Yannick Schreiber, Sandra Trautmann, Benjamin Schnappauf, Formal analysis, Validation, Investigation, Methodology; Saskia Wedel, Formal analysis, Investigation, Visualization, Methodology, Writing – original draft; Dominique Thomas, Nerea Ferreiros Bouzas, Formal analysis, Supervision, Validation, Investigation, Methodology; Christian H Brandts, Resources, Supervision, Funding acquisition, Project administration; Christine Solbach, Resources, Supervision, Project administration; Gerd Geisslinger, Conceptualization, Resources, Supervision, Funding acquisition, Writing – original draft, Project administration; Marco Sisignano, Conceptualization, Formal analysis, Supervision, Funding acquisition, Validation, Investigation, Visualization, Methodology, Writing – original draft, Writing – review and editing

## Author ORCIDs
Marco Sisignano (iD) https://orcid.org/0000-0002-7581-0951

## Ethics
This study was conducted in accordance with the Declaration of Helsinki on Biomedical Research Involving Human Subjects and was approved by the Ethics Committee of the Medical Faculty of the Goethe-University, Frankfurt am Main, Germany (reference number 4/09). Informed written consent was obtained from each of the participants.

Reviewer #1 (Public Review): https://doi.org/10.7554/eLife.91941.3.sa1
Author response https://doi.org/10.7554/eLife.91941.3.sa2

---

# Additional files

## Supplementary files
• Supplementary file 1. Table of patient characteristics of the 31 patients that gave blood samples before and after chemotherapy from the patient cohort.

• Supplementary file 2. Complete list of lipid mediators included in the analyses, separated by group of lipid and detection method.

• Supplementary file 3. Table of patient characteristics of the 28 patients from the second cohort that gave blood samples before and after the sixth cycle of chemotherapy.ld like to thank Drs Tabea Ost.

• MDAR checklist

## Data availability
The patients' informed consent did not cover the publication of their data, even in deidentified form, on a public platform. Researchers who require access to this data for scientific purposes may contact the senior author (Marco.Sisignano@med.uni-frankfurt.de), who will then refer the request to the department's ethics committee. Relevant Python and R code used for data analysis is available at https://github.com/JornLotsch/PaclitaxelNeuropathyProject (copy archived at *Lötsch, 2024*).

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
