## [Editor Report · eLife assessment]

Sisigano et al. report findings about the role of sphingolipids using lipidomics with machine learning in paclitaxel-induced peripheral neuropathy and preliminary translation of the impact of SA1P in cultured neuronal cells. This study presents a **valuable** finding on the increased activity of two well-studied signal transduction pathways in a subtype of breast cancer. The significance is limited by **incomplete** evidence which can be addressed in larger clinical cohorts in the future and with more robust biological validation approaches.

---

## [Referee Report · Reviewer #1 (Public Review)]

Summary:

This study examines lipid profiles in cancer patients treated with the neurotoxic chemotherapy paclitaxel. Multiple methods, including machine learning as well as more conventional statistical modelling, were used to classify lipid patterns before and after paclitaxel treatment and in conjunction with neuropathy status. Lipid profiles before and after paclitaxel therapy were analysed from 31 patients. The study aimed to characterize from the lipid profile if plasma samples were collected pre paclitaxel or post paclitaxel and their relevance to neuropathy status. Sphingolipids including sphinganine-1-phosphate (SA1P) differed between patients with and without neuropathy. To examine the potential role of SA1P, it was applied to murine primary sensory neuron cultures, and produced calcium transients in a proportion of neurons. This response was abolished by application of a TRPV1 antagonist. The number of neurons responding to SA1P was partially reduced by the sphingosine 1-phosphate receptor (S1PR1) modulator fingolimod.

Strengths:

The strengths of this study include the use of multiple methods to classify lipid patterns and the attempt to validate findings from the clinical cohort in a preclinical model using primary sensory neurons.

Weaknesses:

These still stand from the original review and are repeated here:

There are a number of weaknesses in the study. The small sample size is a significant limitation of the study. Out of 31 patients, only 17 patients were reported to develop neuropathy, with significant neuropathy (grade 2/3) in only 5 patients. The authors acknowledge this limitation in the results and discussion sections of the manuscript, but it limits the interpretation of the results. Also acknowledged is the limited method used to assess neuropathy.

Potentially due to this small number of patients with neuropathy, the machine learning algorithms could not distinguish between samples with and without neuropathy. Only selected univariate analyses identified differences in lipid profiles potentially related to neuropathy.

Three sphingolipid mediators including SA1P differed between patients with and without neuropathy at the end of treatment. These sphingolipids were elevated at end of treatment in the cohort with neuropathy, relative to those without neuropathy. However, across all samples from pre to pos- paclitaxel treatment, there was a significant reduction in SA1P levels. It is unclear from the data presented what the underlying mechanism for this result would be. If elevated SA1P is associated with neuropathy development, it would be expected to increase in those who develop neuropathy from pre to post-treatment timepoints.

Primary sensory neuron cultures were used to examine the effects of SA1P application. SA1P application produced calcium transients in a small proportion of sensory neurons. It is not clear how this experimental model assists in validating the role of SA1P in neuropathy development as there is no assessment of sensory neuron damage or other hallmarks of peripheral neuropathy. These results demonstrate that some sensory neurons respond to SA1P and that this activity is linked to TRPV1 receptors. However, further studies will be required to determine if this is mechanistically related to neuropathy.

Impact:

Taken in total, the data presented do not provide sufficient evidence to support the contention that SA1P has an important role in paclitaxel induced peripheral neuropathy. Further, the results do not provide evidence to support the use of S1PR1 receptor antagonists as a therapeutic strategy. It is important to be careful with language use in the discussion, as the significance of the present results are overstated.

However, based on the results of previous studies, it is likely that sphingolipid metabolism plays a role in chemotherapy induced peripheral neuropathy. Based on this existing evidence, the S1PR1 receptor antagonist fingolimod has already been examined in experimental models and in clinical trials. Further work is needed to examine the links between lipid mediators and neuropathy development and identify additional strategies for intervention.

---

## [Author Response]

The following is the authors’ response to the current reviews.

The concerns raised during the review have been incorporated into the discussion of the results, and the need for further research is acknowledged in the paper. This is not possible in the present study, as the clinical project has been completed and further patients cannot be enrolled without starting a new project. We are confident that the results are scientifically valid and that the methodology was scientifically sound and up to date. They were obtained on a dataset that was obviously large enough to allow 20% of it to be set aside and a machine-learned classifier to be trained on the remaining 80%, which then assigned samples to neuropathy with an accuracy better than guessing.

Furthermore, our results are at least tentatively replicated in a completely independent data set from another patient cohort. The strengths and limitations of the study design, in particular the latter, are discussed in the necessary depth. In summary, the machine-learned results provided major hits on one side and probably unimportant lipids on the other side of the variable importance scale. Both could be verified in vitro. We are therefore confident that we have contributed to the advancement of knowledge about cancer therapy-associated neuropathy and look forward to further developments in this area.

The following is the authors’ response to the original reviews.

Weaknesses Reviewer 1:There are a number of weaknesses in the study. The small sample size is a significant limitation of the study. Out of 31 patients, only 17 patients were reported to develop neuropathy, with significant neuropathy (grade 2/3) in only 5 patients. The authors acknowledge this limitation in the results and discussion sections of the manuscript, but it limits the interpretation of the results. Also acknowledged is the limited method used to assess neuropathy.

We agree with the reviewer that the cohort size and assessment of neuropathy are limitations of our study as we already described in the corresponding section of the manuscript. However, occurrence and grade of the neuropathy are in line with results reported from previous studies. From these studies, the expected occurrence of neuropathy with our therapeutic regimen is around 50-70% (54.9% in our cohort), and most patients (80-90%) are expected to experience Grade 1 neuropathy after 12 weeks (13). In these studies, neuropathy is assessed by using questionnaires or by grading via NCTCTCAE as in our study. In summary, assessment and occurrence of neuropathy of our reported cohort are in line with previous reports.

Potentially due to this small number of patients with neuropathy, the machine learning algorithms could not distinguish between samples with and without neuropathy. Only selected univariate analyses identified differences in lipid profiles potentially related to neuropathy.

The data analysis consistently followed a "mixture of experts" approach, as this seems to be the most successful way to deal with omics data. We have elaborated on this in the Methods section, including several supporting references. Regarding the quoted sentence from the results section, after rereading it, we realized that it was somewhat awkwardly worded. What we mean is now better worded in the results section, namely “Although the three algorithms detected neuropathy in new cases, unseen during training, at balanced accuracy of up to 0.75, while only the guess level of 0.5 was achieved when using permuted data for training, the 95% CI of the performance measures was not separated from guess level”. Therefore, multivariate feature selection was not considered a valid approach, since it requires that the algorithms from which the feature importance is read can successfully perform their task of class assignment (4). Therefore, univariate methods (Cohen's d, FPR, FWE) were preferred, as well as a direct hypothesis transfer of the top hits from the abovementioned day1/2 assessments to neuropathy. Classical statistics consisting of direct group comparisons using Kruskal-Wallis tests (5) were performed.”

It was our approach to investigate the data set in an unbiased manner by different machine learning algorithms and select those lipids that the majority of the algorithms considered important for distinguishing the patient groups (majority voting). This way, the inconsistencies and limitations of a single evaluation method, such as regression analysis, that occur in some datasets, can be mitigated.

Three sphingolipid mediators including SA1P differed between patients with and without neuropathy at the end of treatment. These sphingolipids were elevated at the end of treatment in the cohort with neuropathy, relative to those without neuropathy. However, across all samples from pre to post-paclitaxel treatment, there was a significant reduction in SA1P levels. It is unclear from the data presented what the underlying mechanism for this result would be.

We agree with the reviewer that our study does not identify the mechanism by which paclitaxel treatment alters sphingolipid concentrations in the plasma of patients. It has been reported before that paclitaxel may increase expression and activity of serine palmitoyltransferase (SPT) which is the crucial enzyme and rate-limiting step in the *denovo* synthesis of sphingolipids. This may be associated with a shift towards increased synthesis of 1-deoxysphingolipids and a decrease of “classical” sphingolipids (6) and may explain the general reduction of SA1P and other sphingolipid levels after paclitaxel treatment in our study.

It is also conceivable that paclitaxel reduces the release of sphingolipids into the plasma. Paclitaxel is a microtubule stabilizing agent (7) that may interfere with intracellular transport processes and release of paracrine mediators.

The mechanistic details of paclitaxel involvement in sphingolipid metabolism or transport are highly interesting but identifying them is beyond the scope of our manuscript.

If elevated SA1P is associated with neuropathy development, it would be expected to increase in those who develop neuropathy from pre to post-treatment time points.

There is a general trend of reduced plasma SA1P concentrations following paclitaxel treatment. Nevertheless, patients experiencing neuropathy exhibit significantly elevated SA1P levels post-treatment.

It has been shown before that paclitaxel-induced neuropathic pain requires activation of the S1P1 receptor in a preclinical study (8). Moreover, a meta-analysis of genome-wide association studies (GWAS) from two clinical cohorts identified multiple regulatory elements and increased activity of S1PR1 associated with paclitaxel-induced neuropathy (9). These data imply that enhanced S1P receptor activity and signaling are key drivers of paclitaxel-induced neuropathy. It seems that both, increased levels of the sphingolipid ligands in combination with enhanced expression and activity of S1P receptors can potentiate paclitaxel-induced neuropathy in patients. This explains why also decreased SA1P concentrations after paclitaxel treatment can still enhance neuropathy via the S1PRTRPV1 axis in sensory neurons.

We added this paragraph to the discussions section of our manuscript.

Primary sensory neuron cultures were used to examine the effects of SA1P application.SA1P application produced calcium transients in a small proportion of sensory neurons. It is not clear how this experimental model assists in validating the role of SA1P in neuropathy development as there is no assessment of sensory neuron damage or other hallmarks of peripheral neuropathy. These results demonstrate that some sensory neurons respond to SA1P and that this activity is linked to TRPV1 receptors. However, further studies will be required to determine if this is mechanistically related to neuropathy.

As we detected elevated levels of SA1P in the plasma of PIPN patients, we can assume higher concentrations in the vicinity of sensory neurons. These neurons are the main drivers for neuropathy and neuropathic pain and are strongly affected by paclitaxel in their activity (10-15). Also, TRPV1 shows altered activity patterns in response to paclitaxel treatment (16). Because of its relevance for nociception and pathological pain, TRPV1 activity is a suitable and representative readout for pathological pain states in peripheral sensory neurons (17, 18), which is why we investigated them.

We would like to point out the potency of SA1P to increase capsaicin-induced calciumtransients in sensory neurons at submicromolar concentrations.

We also agree with the reviewer that further studies need to investigate the underlying mechanisms in more detail. We added this sentence to the final paragraph in the discussion section of our manuscript.

Weaknesses Reviewer 2:The article is poorly written, hindering a clear understanding of core results. While the study's goals are apparent, the interpretation of sphingolipids, particularly SA1P, as key mediators of paclitaxel-induced neuropathy lacks robust evidence.

We agree that the relevance of SA1P as key mediator of paclitaxel-induced neuropathy might be overstated and changed the wording throughout the manuscript accordingly. However, we would like to point out the potency of this lipid to increase capsaicin-induced calcium-transients in sensory neurons at submicromolar concentrations.

Also, the lipid signature in the plasma of PIPN patients shows a unique pattern and sphingolipids are the group that showed the strongest alterations when comparing the patient groups. We also measured eicosanoids, such as prostaglandins, linoleic acid metabolites, endocannabinoids and other lipid groups that have previously been associated with influences on pain perception or nociceptor sensitization. However, none of these lipids showed significant differences in their concentrations in patient plasma. This is why we consider sphingolipids as contributors to or markers of paclitaxel-induced neuropathy in patients.

We also revised the entire article to improve its clarity.

The introduction fails to establish the significance of general neuropathy or peripheral neuropathy in anticancer drug-treated patients, and crucial details, such as the percentage of patients developing general neuropathy or peripheral neuropathy, are omitted. This omission is particularly relevant given that only around 50% of patients developed neuropathy in this study, primarily of mild Grade 1 severity with negligible symptoms, contradicting the study's assertion of CIPN as a significant side effect.

As we already described in the introduction, CIPN is a serious dose- and therapy-limiting side effect, which affects up to 80% of treated patients. This depends on dose and combination of chemotherapeutic agents. For paclitaxel, therapeutic doses range from 80 – 225 mg/m². As CIPN symptoms are dose-dependent, the number of PIPN patients that receive a high paclitaxel dose is higher than the number of PIPN patient receiving a low dose.

In our study, we mainly used a low dose paclitaxel, because this therapeutic regimen is the most widely used paclitaxel monotherapy. From previous studies, the expected occurrence of neuropathy with this therapeutic regimen is around 50-70%, and most patients (8090%) are expected to experience Grade 1 neuropathy after 12 weeks (1-3).

Our results are within the range reported by these studies (54.9% patients with neuropathy). Also, as we highlight in Table S1, the neuropathy symptoms persist in most cases for several years after chemotherapy, affecting quality of life of these patients which makes it far from being a negligible symptom.

We added some more information concerning PIPN in the introduction section in which we emphasize the clinical problem.

The lack of clarity in distinguishing results obtained by lipidomics using machine learning methods and conventional methods adds to the confusion. The poorly written results section fails to specify SA1P's downregulation or upregulation, and the process of narrowing down to sphingolipids and SA1P is inadequately explained.

We have tried to keep the machine learning part in the main manuscript short and moved major parts of it to a supplement. However, as this has been claimed to have led to a lack of clarity, we have expanded the description of the data analysis and added extensive explanations and supporting references for the mixed expert approach that was used throughout the analysis. We hope this is now clear.

Integrating a significant portion of the discussion section into the results section could enhance clarity. An explanation of the utility of machine learning in classifying patient groups over conventional methods and the citation of original research articles, rather than relying on review articles, may also add clarity to the usefulness of the study.

As suggested by the reviewer, we moved the relevant parts from the discussion to the results section in the revised version of our manuscript.

**Reviewer #1 (Recommendations For The Authors):**
Figure 2 should be better explained or removed. In its current form, it does not add to the interpretation of the manuscript.

As mentioned above, we have expanded the description of the ESOM/U-matrix method in the Methods section and rewritten the figure legend. In addition, we have annotated the U-matrix in the figure. The method has been reported extensively in the computer science and biomedical literature, and a more detailed description in the referenced papers would go beyond the current focus on lipidomics. However, we believe that this discussion is sufficiently detailed for the readers of this report: "… a second unsupervised approach was used to verify the agreement between the lipidomics data structure and the prior classification, implemented as self-organizing maps (SOM) of artificial neurons (19). In the special form of an “emergent” SOM (ESOM (20)), the present map consisted of 4,000 neurons arranged on a two-dimensional toroidal grid with 50 rows and 80 columns (21, 22). ESOM was used because it has been repeatedly shown to correctly detect subgroup structures in biomedical data sets comparable to the present one (20, 22, 23). The core principle of SOM learning is to adjust the weights of neurons based on their proximity to input data points. In this process, the best matching unit (BMU) is identified as the neuron closest to a given data point. The adaptation of the weights is determined by a learning rate (η) and a neighborhood function (h), both of which gradually decrease during the learning process. Finally, the groups are projected onto separate regions of the map. On top of the trained ESOM, the distance structure in the high-dimensional feature space was visualized in the form of a so-called U-matrix (24) which is the canonical tool for displaying the distance structures of input data on ESOM (21).

The visual presentation facilitates data group separation by displaying the distances between BMUs in high-dimensional space in a color-coding that uses a geographical map analogy, where large "heights" represent large distances in feature space, while low "valleys" represent data subsets that are similar. "Mountain ranges" with "snow-covered" heights visually separate the clusters in the data. Further details about ESOM can be found in (24)."

The second patient cohort is only included in the discussion - with cohort details in the supplementary material and figures included in the main text. Perhaps these data should be removed entirely. The findings are described as trends and not statistically significant and multiple issues with this second cohort are mentioned in the discussion.

We agree with the reviewer that including the second patient cohort in the discussion is inadequate. Of course, there are differences between the patient cohorts that do not allow direct comparison and that are highlighted in the section on limitations of the study. However, we still think it is interesting and relevant to show these data, because we used our algorithms trained on the first patient cohort to analyze the second cohort. And these data support the main results.

We therefore moved the entire paragraph to the results section of to improve coherence of our manuscript. The passage was introduced with the subheading: “Support of the main results in an independent second patient cohort”.

The title does not reflect the content of the paper and should be changed to better reflect the content and its significance.

We change the title to “Machine learning and biological validation identify sphingolipids as potential mediators of paclitaxel-induced neuropathy in cancer patients” to avoid overstating the results as suggested by the Reviewer.

Further, the discussion should be modified to avoid overstating the results.

As the reviewer suggests, we changed the wording to avoid overstating the results.

**Reviewer #2 (Recommendations For The Authors):**
Please address the absence of clear neuropathy in the majority of patients after treatment with paclitaxel in your discussion.

As stated above, occurrence and grade of the neuropathy are in line with the results from previous studies. From these studies, the expected occurrence of neuropathy with our therapeutic regimen is around 50-70%, (the variability is due to differences in the assessment methods) and most patients (80-90%) are expected to experience Grade 1 neuropathy after 12 weeks (1-3).

We added this information in the discussion section of the revised manuscript.

Line 65: Kindly replace review articles with original research articles for proper citation.

We replaced the review articles with original publications, focusing on clinical observations. We added the following publications: Jensen *et al.*, Front Neurosci 2020; Chen *et al.*, Neurobiol Aging 2018; Igarashi *et al.,* J Alzheimers Dis. 2011; Kim *et al.,* Oncotarget 2017 as references 17-20 in the revised version of our manuscript.

Line 260: The mention of SA1P is introduced here without prior reference (do not use words like "again", or "see above", if it is not previously mentioned). Adjust the text for coherence.

We agree with the reviewer that the introduction of SA1P in this passage in incoherent. We replaced the sentence in line 260 with:

The small set of lipid mediators emerging from all three methods as informative for neuropathy included the sphingolipid sphinganine-1-phosphate (SA1P), also known as dihydrosphingosine-1-phosphate (DH-S1P)…”

Lines 301-315: Consider relocating several lines from this section to the results section for improved clarity.

We moved the lines 309-312 explaining the algorithm selection and their validation success in the corresponding results section (Lipid mediators informative for assigning postpaclitaxel therapy samples to neuropathy).

Lines 382-396: Move this content to the results section to enhance the organization and coherence of the manuscript.

We moved the entire paragraph to the results section of our manuscript to improve coherence. The passage was introduced with the subheading: “Support of the main results in an independent second patient cohort”.

References

(1) Barginear M, Dueck AC, Allred JB, Bunnell C, Cohen HJ, Freedman RA, et al. Age and the Risk of Paclitaxel-Induced Neuropathy in Women with Early-Stage Breast Cancer (Alliance A151411): Results from 1,881 Patients from Cancer and Leukemia Group B (CALGB) 40101. *Oncologist.* 2019;24(5):617-23.

(2) Mauri D, Kamposioras K, Tsali L, Bristianou M, Valachis A, Karathanasi I, et al. Overall survival benefit for weekly vs. three-weekly taxanes regimens in advanced breast cancer: A metaanalysis. *Cancer Treat Rev.* 2010;36(1):69-74.

(3) Budd GT, Barlow WE, Moore HC, Hobday TJ, Stewart JA, Isaacs C, et al. SWOG S0221: a phase III trial comparing chemotherapy schedules in high-risk early-stage breast cancer. *J Clin Oncol.* 2015;33(1):58-64.

(4) Lötsch J, and Ultsch A. Pitfalls of Using Multinomial Regression Analysis to Identify ClassStructure-Relevant Variables in Biomedical Data Sets: Why a Mixture of Experts (MOE) Approach Is Better. *BioMedInformatics.* 2023;3(4):869-84.

(5) Kruskal WH, and Wallis WA. Use of Ranks in One-Criterion Variance Analysis. *J Am Stat Assoc.* 1952;47(260):583-621.

(6) Kramer R, Bielawski J, Kistner-Griffin E, Othman A, Alecu I, Ernst D, et al. Neurotoxic 1deoxysphingolipids and paclitaxel-induced peripheral neuropathy. *FASEB J.* 2015;29(11):4461-72.

(7) Field JJ, Diaz JF, and Miller JH. The binding sites of microtubule-stabilizing agents. *Chem Biol.* 2013;20(3):301-15.

(8) Janes K, Little JW, Li C, Bryant L, Chen C, Chen Z, et al. The development and maintenance of paclitaxel-induced neuropathic pain require activation of the sphingosine 1-phosphate receptor subtype 1. *J Biol Chem.* 2014;289(30):21082-97.

(9) Chua KC, Xiong C, Ho C, Mushiroda T, Jiang C, Mulkey F, et al. Genomewide Meta-Analysis Validates a Role for S1PR1 in Microtubule Targeting Agent-Induced Sensory Peripheral Neuropathy. *Clin Pharmacol Ther.* 2020;108(3):625-34.

(10) Kawakami K, Chiba T, Katagiri N, Saduka M, Abe K, Utsunomiya I, et al. Paclitaxel increases high voltage-dependent calcium channel current in dorsal root ganglion neurons of the rat. *J Pharmacol Sci.* 2012;120(3):187-95.

(11) Pittman SK, Gracias NG, Vasko MR, and Fehrenbacher JC. Paclitaxel alters the evoked release of calcitonin gene-related peptide from rat sensory neurons in culture. *Exp Neurol.* 2013.

(12) Luo H, Liu HZ, Zhang WW, Matsuda M, Lv N, Chen G, et al. Interleukin-17 Regulates NeuronGlial Communications, Synaptic Transmission, and Neuropathic Pain after Chemotherapy.

*Cell reports.* 2019;29(8):2384-97 e5.

(13) Pease-Raissi SE, Pazyra-Murphy MF, Li Y, Wachter F, Fukuda Y, Fenstermacher SJ, et al. Paclitaxel Reduces Axonal Bclw to Initiate IP3R1-Dependent Axon Degeneration. *Neuron.* 2017;96(2):373-86 e6.

(14) Duggett NA, Griffiths LA, and Flatters SJL. Paclitaxel-induced painful neuropathy is associated with changes in mitochondrial bioenergetics, glycolysis, and an energy deficit in dorsal root ganglia neurons. *Pain.* 2017.

(15) Li Y, Adamek P, Zhang H, Tatsui CE, Rhines LD, Mrozkova P, et al. The Cancer Chemotherapeutic Paclitaxel Increases Human and Rodent Sensory Neuron Responses to TRPV1 by Activation of TLR4. *J Neurosci.* 2015;35(39):13487-500.

(16) Hara T, Chiba T, Abe K, Makabe A, Ikeno S, Kawakami K, et al. Effect of paclitaxel on transient receptor potential vanilloid 1 in rat dorsal root ganglion. *Pain.* 2013;154(6):882-9.

(17) Jardin I, Lopez JJ, Diez R, Sanchez-Collado J, Cantonero C, Albarran L, et al. TRPs in Pain Sensation. *Front Physiol.* 2017;8:392.

(18) Julius D. TRP Channels and Pain. *Annual review of cell and developmental biology.*

2013;29:355-84.

(19) Kohonen T. Self-Organized Formation of Topologically Correct Feature Maps. *Biol Cybern.* 1982;43(1):59-69.

(20) Lötsch J, Lerch F, Djaldetti R, Tegder I, and Ultsch A. Identification of disease-distinct complex biomarker patterns by means of unsupervised machine-learning using an interactive R toolbox (Umatrix). *Big Data Analytics.* 2018;3(1):5.

(21) Ultsch A. 2003.

(22) Lotsch J, Geisslinger G, Heinemann S, Lerch F, Oertel BG, and Ultsch A. Quantitative sensory testing response patterns to capsaicin- and ultraviolet-B-induced local skin hypersensitization in healthy subjects: a machine-learned analysis. *Pain.* 2018;159(1):11-24.

(23) Lötsch J, Thrun M, Lerch F, Brunkhorst R, Schiffmann S, Thomas D, et al. Machine-Learned Data Structures of Lipid Marker Serum Concentrations in Multiple Sclerosis Patients Differ from Those in Healthy Subjects. *Int J Mol Sci.* 2017;18(6).

(24) Lötsch J, and Ultsch A. Cham: Springer International Publishing; 2014:249-57.